# CoCoQuant: Breaking the Bandwidth Wall via Co-Optimized Communication and Computation Quantization

Haojie Duanmu [1 2 3]   Jifeng Ding [4 3]   Size Zheng [2]   Xuegui Zheng [2]   Jiangfei Duan [5]   Xingcheng Zhang [3]
Li-wen Chang [2]   Xin Liu [2]   Dahua Lin [5]

## Abstract

The rapid scaling of large language models (LLMs) has made distributed inference indispensable, yet end-to-end latency is increasingly dominated by communication, forming a critical bandwidth wall that fundamentally limits the practical gains of existing quantization techniques. Existing approaches typically treat communication and computation in isolation, failing to exploit their coupled nature and introducing limited system-level acceleration and accuracy degradation. To address this, we propose CoCoQuant, a co-designed framework that jointly optimizes communication and computation as a unified end-to-end design space. CoCoQuant introduces a precision-aligned graph-rewriting that enables zero-overhead fusion between low-precision communication and computation. CoCoQuant formulates a hardware-aware mixed-precision allocation problem that integrates roofline-based cost modeling with relative sensitivity calibration, solved via global integer linear programming. Extensive experiments on LLMs of varying scales demonstrate that CoCoQuant achieves Pareto-optimal accuracy-latency trade-offs, delivering up to 2.92 end-to-end speedup with a negligible increase in perplexity (0.22).

## 1. Introduction

State-of-the-art large language model (LLM) serving stacks increasingly rely on distributed inference over multi-

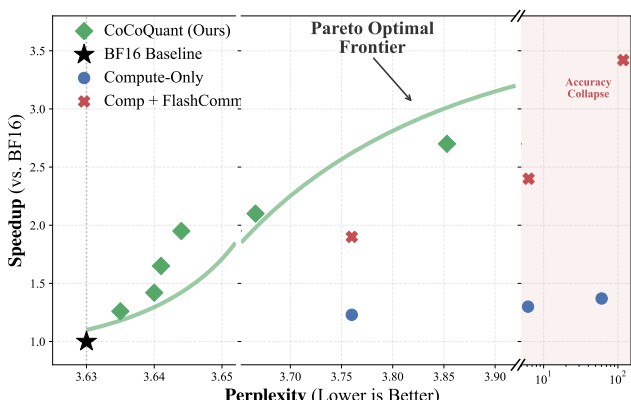

*Figure 1.* End-to-end inference speedup vs. perplexity (lower is better) on Llama-3-70B on 8xRTX-4090. **Compute-Only Quantization** plateaus in speedup due to the communication bottleneck. **Communication Quantization** (i.e. FlashComm) with compute quantization pierces the wall but suffers catastrophic accuracy collapse. Our proposed **CoCoQuant** co-optimizes both dimensions, establishing a superior Pareto frontier.

GPU clusters, where tensor model parallelism shards each layer across devices. In these deployments, end-to-end latency is no longer dominated by computation alone. As model size and context length grow, inter-device collectives such as `AllReduce`, `AllGather`, and `All-to-All` begin to dominate overall latency. At the same time, hardware evolves asymmetrically: while GPU compute throughput continues to scale via tensor cores' native support for low-precision arithmetic, interconnect bandwidth improves much more slowly. This imbalance leads to a *bandwidth wall*, especially on bandwidth-constrained accelerators such as H800 or L20 GPUs, where wall-clock latency is increasingly limited by communication rather than FLOPs. Consequently, compute-side acceleration alone quickly reaches its limit. (Jangda et al., 2022; Li et al., 2024; Chang et al., 2024).

While model quantization has become a cornerstone for reducing memory footprints and accelerating computation, it largely remains confined to a single-device perspective, leaving the emerging distributed bandwidth wall fundamentally unaddressed. To mitigate communication bottle-

[1]Shanghai Jiao Tong University [2]ByteDance Seed [3]Shanghai AI Laboratory [4]Fudan University [5]The Chinese University of Hong Kong. Correspondence to: Size Zheng <zheng.size@bytedance.com>, Haojie Duanmu <duanmuhaojie@sjtu.edu.cn>.

*Proceedings of the 43rd International Conference on Machine Learning*, Seoul, South Korea. PMLR 306, 2026. Copyright 2026 by the author(s).

necks, recent works have begun to explore compression for collective communication operators. However, these two lines of research have evolved in isolation, forming a fragmented optimization paradigm. Specifically, computation-centric quantization (Dettmers et al., 2022; Xiao et al., 2023; Ashkboos et al., 2024; Zhang et al., 2025) focus on exploiting low-precision arithmetic units within accelerators, while communication-centric approaches (Wang et al., 2023; Li et al., 2024; 2025; Zhao et al., 2025) compress collective communications independently, treating the surrounding computational graph as an opaque boundary.

This disjoint design manifests structurally as a *fragmented precision flow*, leading to critical inefficiencies. First, rigid precision boundaries introduce redundant quantization-dequantization (Q/DQ) overheads, degrading both accuracy and computational efficiency. Moreover, these boundaries preclude system-level optimization by preventing cross-operator fusion. Consequently, as shown in Figure 1, the naive composition of computation quantization and communication compression fails to translate theoretical efficiency gains into proportional wall-clock speedups, yielding suboptimal accuracy-efficiency trade-offs. Beyond these structural fragmentation issues, navigating the configuration space presents a distinct challenge: the optimal precision allocation depends jointly on the hardware's compute-to-bandwidth ratio and the varying quantization sensitivity of different operators (Appendix A). Static, hand-tuned policies are therefore fundamentally inadequate for diverse deployment scenarios.

Together, the enormous potential gains from breaking the bandwidth wall and the fundamental limitations of existing fragmented designs raise a central question: **Can we systematically co-design computation and communication quantization as a unified, hardware-aware optimization problem and solve it automatically?**

To this end, we propose CoCoQuant, a co-designed framework that unifies collective communication and computation into a holistic quantization design space for tensor-parallel LLM inference. Instead of treating quantization as an isolated operator-level transformation, CoCoQuant introduces a precision-aligned graph rewriting mechanism to enforce precision consistency across computation-communication boundaries, eliminating Q/DQ between the two stages. As a result, fragmented precision flows are transformed into a unified low-precision dataflow, where precision alignment becomes a structural property of the execution graph rather than a local operator decision. Building on this optimized execution structure, CoCoQuant employs hardware-aware bitwidth allocation for mixed-precision execution, automatically navigating the complex trade-off between model accuracy, compute throughput, and interconnect efficiency. We formulate the mixed-

precision bit-width allocation as an $\epsilon$-constraint Integer Linear Programming (ILP) problem, solving for the optimal quantization strategy specific to the target hardware configuration. Furthermore, to translate the theoretical optimality into wall-clock reduction, CoCoQuant provides a suite of fused kernel templates that automatically compose to support arbitrary mixed-precision configurations, fully hiding the complexity of mixed-quantization management from users.

Empirical results demonstrate that CoCoQuant delivers up to **2.92×** end-to-end speedup with negligible accuracy degradation (keeping perplexity increase within **0.22**), consistently establishing a superior Pareto frontier over state-of-the-art computation-only and communication-only baselines.

## 2. Background and Motivation

### 2.1. Quantization Fundamentals

Quantization reduces numerical precision to alleviate memory footprint and bandwidth pressure while enabling high-throughput low-precision arithmetic. We focus on *uniform quantization*, which maps a high-precision vector $\mathbf{x}$ to discrete integer levels within a range $[q_{\min}, q_{\max}]$ determined by the bit-width $b$. The quantization operator $\mathcal{Q}(\cdot)$ and dequantization operator $\mathcal{D}(\cdot)$ are formally defined as:

$$\mathbf{x}_q = \mathcal{Q}(\mathbf{x}) = \text{clamp}\left(\left\lfloor\frac{\mathbf{x}}{s}\right\rceil + z,\ q_{\min},\ q_{\max}\right), \quad (1)$$

$$\hat{\mathbf{x}} = \mathcal{D}(\mathbf{x}_q) = s \cdot (\mathbf{x}_q - z), \quad (2)$$

where $\lfloor\cdot\rceil$ denotes rounding operation. The scaling factor $s \in \mathbb{R}^+$ and zero-point $z \in \mathbb{Z}$ are parameters derived from the data distribution. This formulation unifies two common modes: asymmetric quantization, which utilizes a calculated $z$ to capture unbalanced distributions, and symmetric quantization, which enforces $z = 0$ to reduce computational overhead at runtime.

Quantization strategies generally fall into two categories depending on which tensors are compressed. *Weight-only quantization* compresses strictly model weights to address memory-bound bottlenecks (e.g., during decoding) but requires on-the-fly dequantization before computation (Frantar et al., 2022; Kim et al., 2023; Lin et al., 2024). Conversely, *weight-activation quantization* compresses both weights and input tokens, enabling end-to-end execution on low-precision units (e.g., INT8 or FP8 Tensor Cores) (Dettmers et al., 2022; Xiao et al., 2023; Ashkboos et al., 2024). In this work, we prioritize weight-activation quantization to simultaneously exploit memory bandwidth savings and computational acceleration.

Mixed precision quantization allocates different bit-widths to different parts of the model based on certain proxy,

which has been shown to enhance model accuracy or accuracy-efficiency tradeoff (Dong et al., 2019; Wang et al., 2019; Yao et al., 2021; Zhao et al., 2024; Duanmu et al., 2025). Our study falls into this category but the first to co-optimize communication and computation instead of limited on single device.

## 2.2. Distributed Inference and Bandwidth Wall

As LLMs scale to hundreds of billions of parameters, single-device deployment becomes infeasible due to the dual constraints of memory capacity and computational throughput. For instance, the DeepSeek-V3 model (671B parameters) exceeds the aggregate memory capacity of an $8\times$H100 node even under FP8 quantization (Liu et al., 2024). Furthermore, latency-sensitive applications impose stringent Time-To-First-Token (TTFT) constraints, making distributed inference necessary not only for memory scalability but also for meeting real-time service-level objectives (SLOs) through parallel execution (Patel et al., 2024; Zhong et al., 2024).

To manage this scale, standard parallelization strategies partition the model across accelerators. **Tensor Parallelism (TP)** splits individual weight matrices, necessitating two blocking synchronization steps via `AllReduce` in every Transformer layer. In contrast, **Expert Parallelism (EP)**, designed for Mixture-of-Experts (MoE) models, distributes experts and utilizes `All-to-All` communication for token routing. While EP demonstrates superior scalability at massive cluster sizes, we prioritize TP in this work as it remains the foundational strategy for both dense and MoE models. Moreover, at practical deployment scales (e.g., intra-node or moderate-scale clusters), TP is often preferred to avoid the dynamic load imbalance inherent in EP and to maintain deterministic latency characteristics (Jin et al., 2025; Zhu et al., 2025).

However, the efficacy of TP is increasingly constrained by the disparity between compute and communication capabilities. We formally define this phenomenon as the **Bandwidth Wall**: *a structural bottleneck where the growth of arithmetic throughput significantly outpaces that of interconnect bandwidth, causing communication to dominate end-to-end performance.* As standard interconnects lag behind accelerator compute power, the collective `AllReduce` operations in TP inference become the critical path. As shown in Table 1, this wall is particularly severe on bandwidth-constrained hardware (e.g., H800, consumer-grade GPUs), where accelerators spend a substantial portion of cycles idling for data exchange.

*Table 1.* Latency breakdown of Llama-3-70B tensor parallel inference using vLLM (Kwon et al., 2023) across hardware platforms. Sequence length is 8K, batch size is 4, and tensor parallel degree is 8. **BW** denotes unidirectional interconnect bandwidth. **Comp.** and **Comm.** denote the proportions of local computation and cross-device communication time. NVLink v4[*] indicates bandwidth-capped NVLink.

| Hardware | Interconnect | BW (GB/s) | Comp. (%) | Comm. (%) |
|---|---|---|---|---|
| H100 | NVLink v4 | 450 | 69.4 | 30.6 |
| H800 | NVLink v4[*] | 200 | 48.7 | 51.3 |
| RTX 4090 | PCIe Gen4 x16 | 32 | 40.7 | 59.3 |

## 2.3. Communication Compression with Fragmented Precision Flow

To mitigate the bandwidth wall, recent works such as Flash-Comm (Li et al., 2024; 2025) have proposed communication compression. These methods typically decompose `AllReduce` into an `All-to-All`, a local reduce sum and an `AllGather`, applying block-wise quantization to the communicated data to reduce transfer volume. While effective in isolation, these approaches typically treat communication compression as a standalone module, detached from the computational precision context. This modularity leads to a fragmented precision flow, characterized by three key inefficiencies: ❶**Redundant quantization-dequantization overhead**. Figure 2 demonstrate in a typical pipeline where the communication and subsequent computation are both quantized, the lack of coordination forces a "ping-pong" precision conversion. The quantized communication operator is forced to dequantize data back to full-precision for subsequent residual add, RM-SNorm, MoE Routing etc., only for the system to re-quantize the data immediately for the next computation. These redundant operations are memory-bound, consuming valuable DRAM bandwidth and offsetting the latency gains achieved by compressing the communication itself. ❷**Compounding Quantization Error.** Each Q/DQ pair introduces independent quantization noise. In a fragmented precision flow with five such operations (as shown in Figure 2), these errors propagate and compound along the critical path, for example, MoE router, which is sensitive to quantization error (Fu et al., 2025). The activation tensor undergoes repeated lossy transformations, accumulating distortion that degrades model fidelity far beyond what any single quantization stage would suggest. This error amplification is particularly severe for aggressive low-precision configurations (e.g., INT4). ❸**Missed Fusion Opportunities.** To achieve efficiency, quantized operators fuse Q/DQ operations inside the quantized `AllReduce` together with communication in one operator (Li et al., 2024). While

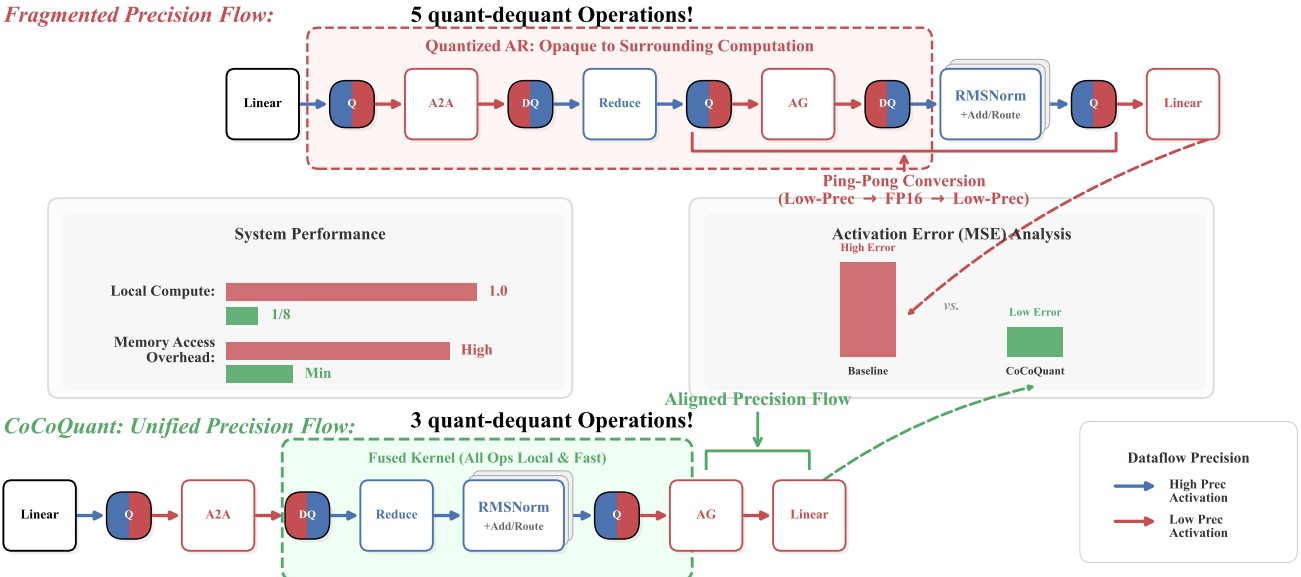

Figure 2. **Structural comparison of precision dataflows in Tensor Parallelism. Top (Baseline):** The *Fragmented Precision Flow* treats quantized communication (red dashed box) as an opaque kernel. This isolation forces a **"Ping-Pong Conversion"** to accommodate row-wise operations (e.g., RMSNorm, MoE Gate), incurring redundant cast overheads (5 ops) and accumulating quantization error (High MSE). **Bottom (CoCoQuant):** We propose a *Unified Precision Flow* via graph rewriting. By fusing row-wise operators into a local kernel (green dashed box) before communication, CoCoQuant achieves an **aligned precision flow**. This allows the AllGather output to flow directly into the subsequent Linear layer, reducing quantization-dequantization steps from 5 to 3 and significantly lowering both system latency and activation error.

such intra-operator fusion makes single quantized communication operator efficient, it also makes such operator an opaque black-box to surrounding computations, making the system hard to perform more aggressive operator fusion, leading to sub-optimal end-to-end efficiency.

This analysis underscores that treating communication compression and computation quantization as orthogonal problems is insufficient. To break the bandwidth wall effectively, a unified design that enforces precision alignment across operator boundaries is required.

## 3. CoCoQuant

In this section, we introduce CoCoQuant, a framework that co-optimizes communication and computation quantization for tensor-parallel LLM inference. CoCoQuant explicitly models the coupled nature of computation and communication, and jointly optimizes them as a unified end-to-end problem. CoCoQuant comprises three key components: (1) *precision-aligned graph rewriting* that enables precision consistency across operator boundaries, (2) *hardware-aware mixed-precision allocation* via integer linear programming that navigates the accuracy-latency trade-off, and (3) *fused kernel templates* that translate the optimized configuration into efficient execution.

### 3.1. Precision-Aligned Graph Rewriting

Existing distributed inference systems typically treat collective communication as an opaque barrier, performing quantization and dequantization strictly at operator boundaries. To break this isolation, we introduce a **Precision-Aligned Graph Rewriting** mechanism that restructures the execution topology of Tensor Parallelism.

As illustrated in Figure 2, CoCoQuant propels element-wise and row-wise operations (e.g., RMSNorm, Residual Add, MoE Gating) from the *replicated phase* (post-AllGather) upstream to the *sharded phase* (pre-AllGather):

$$f(\text{AllGather}(\mathbf{x})) \equiv \text{AllGather}(f(\mathbf{x})). \quad (3)$$

The equivalence holds for operators that act independently on each token or row and do not mix information across the gathered dimension. This class covers the operators that appear on the TP critical path in mainstream Transformer blocks, including RMSNorm, residual addition, element-wise activation functions, and per-token MoE routing. It does not apply to operators with cross-token dependencies; those operators must remain outside this rewrite unless their communication pattern is changed accordingly. By exploiting this property, we push $f(\cdot)$ into the local computation scope before the collective primitive.

This graph rewriting yields three critical advantages:

❶**Reduced Local Computation:** By shifting operations to the sharded phase, element-wise kernels process data chunks of size $1/$TP compared to the full sequence length in the replicated phase. This effectively reduces the memory access of these memory-bound operators by a factor of the tensor parallel degree (TP). ❷**Precision Protection:** Recent researches indicate that RMSNorm and MoE router are highly sensitive to precision (Fu et al., 2025). In this rewrote graph, these operations are fused immediately after the local `Reduce-Sum`. This allows them to consume high-precision (FP32/BF16) accumulation results held in registers or L1 cache, insulating sensitive statistics from quantization noise before they are compressed for the subsequent `AllGather`. ❸**Unified Precision Flow:** This rewriting clears the quantization barriers on the critical path. By ensuring the `AllGather` outputs data in a format directly consumable by the subsequent GEMM, we eliminate redundant dequantization steps, reducing the number of cast operations from 5 to 3 per layer (as shown in Figure 2). Furthermore, such uniform precision flow in communication and computation creates an opportunity of exploit tile-level overlapping (Sul et al., 2025; Chang et al., 2024; Zheng et al., 2025a;b) and we leave this in future work.

For a transformer layer, we use this graph rewriting mechanism to enforce isomorphic precision constraints for all the communication-computation pairs. Specifically, let $q_{\mathrm{AG}}^{(l)}$ denote the output quantization configuration of the `AllGather` at layer $l$, and $q_{\mathrm{act}}^{(l,m)}$ denote the input activation quantization required by module $m$. We impose the following constraints:

$$q_{\mathrm{AG}}^{(l,\mathrm{attn})} \equiv q_{\mathrm{act}}^{(l,\mathrm{ffn}_1)} \qquad (4)$$

$$q_{\mathrm{AG}}^{(l-1,\mathrm{ffn}_2)} \equiv q_{\mathrm{act}}^{(l,\mathrm{attn}_{\mathrm{qkv}})} \qquad (5)$$

Constraint (4) aligns the Attention block's communication output with the FFN's first Linear layer, while Constraint (5) aligns the FFN's output with the next layer's QKV projection. These constraints effectively "short-circuit" the high-precision bubbles found in standard pipelines.

The proposed graph rewriting necessitates synchronizing auxiliary metadata alongside the primary tensor data during `AllGather`. This includes quantization scales for the quantized data and, for Mixture-of-Experts (MoE) models, the routing indices/weights generated by the gating function. We implement this by appending the metadata to the communication payload. Empirical analysis shows that this overhead is negligible: depending on the model architecture, the metadata typically occupies only 50 to 200 bytes per token. Compared to the kilobytes of activation tensors transferred in large-scale models (e.g. activation size for one token of Llama3.3-70B is 16KB), this minimal band-width consumption has virtually no impact on end-to-end latency.

### 3.2. Hardware-Aware Mixed-Precision Allocation

With precision alignment enabled by graph rewriting, we now formulate the bitwidth allocation as an optimization problem. The goal is to find the optimal quantization configuration for each module that minimizes accuracy degradation while satisfying hardware latency constraints.

We decompose each Transformer layer into eight quantizable modules:

$$\mathcal{M}_{comp} = \{\mathrm{attn}_{\mathrm{qkv}}, \mathrm{attn}_{\mathrm{o}}, \mathrm{ffn}_1, \mathrm{ffn}_2\}$$
$$\mathcal{M}_{comm} = \{\mathrm{attn}_{\mathrm{rs}}, \mathrm{attn}_{\mathrm{ag}}, \mathrm{ffn}_{\mathrm{rs}}, \mathrm{ffn}_{\mathrm{ag}}\}$$

Each module can be assigned a quantization configuration from a candidate set $\mathcal{Q}$, which includes various bit-widths (4-bit, 8-bit, FP8, FP16) and group sizes.

**Sensitivity Calibration.** To quantify the accuracy impact of each configuration, we employ relative Frobenius norm as the metric of quantization sensitivity. For module $m$ at layer $l$ under configuration $q$, the quantization loss is quantified:

$$D_{l,m,q} = \frac{\|\hat{\mathbf{Y}}_{l,m,q} - \mathbf{Y}_{l,m}\|_F}{\|\mathbf{Y}_{l,m}\|_F + \epsilon} \qquad (6)$$

where $\mathbf{Y}_{l,m}$ is the full-precision output of layer $l$ and $\hat{\mathbf{Y}}_{l,m,q}$ is the output of layer $l$ with module $m$ applied quantization strategy $q$, $\epsilon$ is a small positive constant for numerical stability to avoid division by zero. This relative metric normalizes the error by signal energy, providing scale-invariant sensitivity estimates across layers with varying activation magnitudes. A series of works have explored the relationship between the quantization error of individual modules and the model (Choukroun et al., 2019; Dong et al., 2019; Frantar et al., 2022). In this study, we adopt the setting presented in (Choukroun et al., 2019) which assumes a positive correlation between the change in the intermediate output of the quantized layer and the final output. Therefore, minimizing the intermediate output loss leads to minimize the layer-wise loss, thus model-wise loss:

$$D_l = \sum_{l \in \mathcal{L}} \sum_{m \in \mathcal{M}} \sum_{q \in \mathcal{Q}} x_{l,m,q} \cdot \frac{\|\hat{\mathbf{Y}}_{l,m,q} - \mathbf{Y}_{l,m}\|_F}{\|\mathbf{Y}_{l,m}\|_F + \epsilon} \qquad (7)$$

In practice, we use a small calibration set (e.g., 128 samples from WikiText2) to get the module-configuration sensitivity statistics.

**Hardware Efficiency Modeling.** We construct the latency cost $T(l, m, q)$ using a hardware roofline model (Williams et al., 2009). For compute-intensive modules (e.g.

GEMM), latency is determined by the arithmetic intensity and hardware throughput:

$$T_{\text{comp}}(l, m, q) = \frac{\text{FLOPs}}{\text{TFLOPS}_q \cdot \eta_{\text{TC}}} \quad (8)$$

where FLOPs is the number of Multiply-And-Add (MAC) operations and $\eta_{\text{TC}}$ is the tensor core efficiency determined by arithmetic intensity of the input shape. For example, for GEMM operation, as the input sequence length is big enough, it falls into compute-bound region and $\eta_{\text{TC}}$ equals to 1, while in short sequence length scenario, $\eta_{\text{TC}}$ is less than 1. For memory bound operations like RMSNorm and Q/DQ, we directly use HBM bandwidth to divide the memory traffic. For communication modules:

$$T_{\text{comm}}(l, m, q) = \frac{\text{DataSize} \cdot b_q/16}{\text{BW}_{\text{link}} \cdot \eta_{\text{comm}}} \quad (9)$$

where $b_q$ is the bit-width of configuration $q$ and $\eta_{\text{comm}}$ is the communication efficiency. For fused communication-computation pairs satisfying the precision alignment constraints, we remove the quant-dequant overhead from the cost model.

$\epsilon$**-Constraint ILP Formulation.** We formulate the allocation as an Integer Linear Programming (ILP) problem using the $\epsilon$-constraint method. Let $x_{l,m,q} \in \{0,1\}$ be binary decision variable indicates whether module $m$ at layer $l$ uses configuration $q$. The optimization problem is:

$$\underset{\mathbf{x}}{\text{minimize}} \quad \sum_{l \in \mathcal{L}} \sum_{m \in \mathcal{M}} \sum_{q \in \mathcal{Q}} x_{l,m,q} \cdot D_{l,m,q} \quad (10)$$

$$\text{subject to} \quad \sum_{l,m,q} x_{l,m,q} \cdot T(l, m, q) \leq T_{\text{budget}} \quad (11)$$

$$\sum_{l,m,q} x_{l,m,q} \cdot M(l, m, q) \leq M_{\text{budget}} \quad (12)$$

$$\sum_{q \in \mathcal{Q}} x_{l,m,q} = 1, \quad \forall l, m \quad (13)$$

$$x_{l,\text{ffn}_1,q} = x_{l,\text{attn}_{\text{ag}},\phi(q)}, \quad \forall l \quad (14)$$

$$x_{l,\text{attn}_{\text{qkv}},q} = x_{l-1,\text{ffn}_{\text{ag}},\phi(q)}, \quad \forall l > 0 \quad (15)$$

Constraint (11) enforces the hardware latency budget. Constraint (12) enforces the compressed model size. Constraint (13) ensures each module has exactly one configuration. Constraints (14) and (15) enforce the precision alignment from subsection 3.1, where $\phi(q)$ maps a weight-activation config to its corresponding activation-only config (e.g., `w4a4_g128 ↦ i4_g128`). Importantly, when these fusion constraints are active, we exclude the communication modules `attn_ag` and `ffn_ag` from the accuracy objective (10), as their quantization error is already accounted for in the fused computation.

By sweeping $T_{\text{budget}}$ from tight (aggressive quantization) to relaxed (conservative), CoCoQuant constructs an exact Pareto frontier of accuracy–latency trade-offs, as illustrated in Figure 1. Given a target model and platform and latency budget, CoCoQuant automatically identifies system bottlenecks and performs adaptive compression. In practice, we first profile the full-precision model latency to establish a baseline. We then define $T_{budget}$ as a fraction of this baseline (e.g. $0.6 \times T_{FP}$) and use `pulp` to solve this problem. This abstraction decouples the configuration search from hardware specifics, allowing users to intuitively select budgets based on their required relative speedup.

### 3.3. Fused Kernel

Breaking fragmented precision flow boundaries brings opportunities for further kernel fusion, but mixed precision also introduces some complexity: we need to properly handle different combinations of packed data formats and hiding the quant-dequant latency. To translate the optimized precision allocation into real-world gains, we implement a suite of fused kernel templates. As shown in the green dashed box of Figure 2, we fuse the following operations into a single kernel:

$$\text{DQ} \rightarrow \text{Reduce} \rightarrow \text{RMSNorm/Add/Route} \rightarrow \text{Q} \quad (16)$$

All intermediate results reside in registers/L1 Cache, eliminating global memory round-trips. The Kernel templates support arbitrary combinations of mixed precision without runtime overhead. With hand-optimized 2000+ lines of CUDA code, these templates support up to 2000+ mixed precision combinations (Appendix C).

## 4. Evaluation

### 4.1. Experimental Setup

**Models.** We evaluate on five representative LLMs spanning different scales and architectures: Llama-3-8B-Instruct, Qwen3-32B, Llama-3.3-70B-Instruct, and Llama4-Scout-109B (MoE), Qwen3-MoE-235B. This selection covers the spectrum from moderate-scale dense models to large-scale MoE architectures.

**Calibration.** CoCoQuant requires offline calibration to determine the quantization sensitivity of computation and communication for bitwidth allocation. For all experiments, we use 128 sequences, each of length 4096, drawn from the Wikitext2 training set (Merity et al., 2016). This calibration process typically takes from several minutes to a few hours depending on the model size.

**Quantization.** To enhance quantization accuracy, we first apply random hadamard transformation (Ashkboos et al., 2024) and then employ GPTQ (Frantar et al., 2022) to quantize model weight for all settings (including all the baseline). For GPTQ, we use the same calibration set with

that of bitwidth allocation.

**System and Baselines.** All end-to-end latency numbers are measured in vLLM (commit `dc837bc23e`). We compare against three classes of baselines in the same stack: (1) full-precision vLLM TP/SP baselines, (2) compute-only quantization baselines such as `W8A8C16` and `W4A4C16`, and (3) uniform communication-computation quantization baselines such as `W8A8C8` and `W4A4C4`. Since vLLM does not natively support all low-bit TP configurations used in our study, we implement the required CUT-LASS per-token/per-channel GEMMs and use optimized group-wise kernels for `W4A4` settings. Importantly, the communication-quantized uniform baselines use the same fused boundary kernels as CoCoQuant wherever applicable. Thus, the reported gains are not obtained by comparing an optimized CoCoQuant implementation against a naive unfused baseline.

### 4.2. Main Results

We evaluate CoCoQuant on several 7 zero-shot tasks and the perplexity of WikiText2 as shown in Table 2. Following FlashComm (Li et al., 2024), we show the end-to-end prefill latency (i.e., TTFT) on a single node with $8\times$RTX 4090 GPUs connected via PCIe Gen4 x16 (32 GB/s per direction). We set TP size uniformly to 8 for all configurations. **CoCoQuant** [1] indicates aggressive budget: i.e. $0.4 \times T_{FP}$ while **CoCoQuant** [2] is a more relaxed budget to get lossless compression used widely in real-world.

**Extra Accuracy and Latency Results.** We also evaluate on reasoning tasks and presents the results in Appendix B. Furthermore, we include latency test results on H800 with NVLink with varies TP size and varies batch size in Appendix D.

**1. Compute-Only Quantization hits the Bandwidth Wall.** Consistent with our motivation, quantizing only computations (`W8A8C16` and `W4A4C16`) yields marginal speedups across all models, typically ranging from $1.1\times$ to $1.3\times$. For instance, on Llama-3-70B, reducing computation precision to 4-bit (`W4A4C16`) only improves speedup to $1.31\times$ despite a significant PPL degradation ($3.63 \rightarrow 7.30$). This empirically confirms that on bandwidth-constrained clusters (e.g., $8\times4090$), end-to-end latency is dominated by communication overhead, rendering compute-centric optimization ineffective.

**2. CoCoQuant establishes a superior Pareto frontier.** Comparing CoCoQuant with naive communication quantization baselines (`W8A8C8`, `W8A8C4`, `W4A4C4*`), our method consistently achieves better accuracy-efficiency trade-offs.

*Higher Speed at Iso-Accuracy:* On Llama-3-70B, CoCoQuant [2] achieves $2.09\times$ speedup with a PPL of 3.66, ef-

fectively matching the full-precision baseline (3.63) while outperforming the conservative `W8A8C8` baseline ($1.82\times$). *Higher Accuracy at Iso-Speed:* On Llama-3-8B, CoCoQuant [1] reaches a speedup of $2.73\times$, comparable to the aggressive `W4A4C4*` ($2.78\times$), but reduces perplexity dramatically from 9.21 to 7.87, recovering most of the accuracy loss.

**3. Robustness and Scalability on MoE Architectures.** The advantages of CoCoQuant consistently generalize from moderate-scale dense models to large-scale MoE architectures (i.e., Llama4-Scout-109B, Qwen3-MoE-235B). Standard low-precision methods often lead to catastrophic failure on MoEs (e.g., `W4A4C16` on Llama4-Scout causes PPL to explode to 457.96). In contrast, CoCoQuant demonstrates exceptional robustness. On Qwen3-MoE-235B, CoCoQuant[1] delivers a remarkable $\mathbf{2.92\times}$ speedup—significantly higher than the aggressive `W4A4C4*` ($2.01\times$)—while maintaining a PPL of 4.18, which is competitive with the FP16 baseline (4.04). This validates that our co-design strategy effectively identifies and protects sensitive expert routing and aggregation operations.

### 4.3. Additional System Analysis

**Fine-grained latency breakdown.** To clarify where the end-to-end speedup comes from without adding a large table to the main text, we provide a stacked prefill breakdown in Appendix E, including Figure 5 and the corresponding absolute numbers in Table 10. On Qwen3-32B with TP8, batch size 4, and average prompt length around 8K tokens on GovReport, CoCoQuant reduces TTFT from 3768.26 ms for the uniform `vLLM SP W8A8C8` baseline to 2889.59 ms. The reduction is not due to comparing against an unfused baseline: the `CoCoQuant w/o Fused` variant has the same A2A+Reduce+AG communication pattern but loses the precision-aligned fused path, causing the residual row-wise and Q/DQ overhead to rise to 1088.77 ms. This confirms that the lower latency comes from jointly selecting lower precision for robust modules and eliminating fragmented precision boundaries.

**Layer-wise allocation patterns.** To make the ILP allocation interpretable, we visualize the selected bit-widths in Figure 6 and provide the detailed discussion in Appendix F. Compared with the uniform `W8A8C4` baseline, CoCoQuant does not simply lower all communication precision. Both the matched-latency CoCoQuant [*] point and the paper configuration preserve higher precision on sensitive early/late communication boundaries and selected FFN modules, while assigning lower precision to more robust middle-layer modules. This pattern explains why mixed precision recovers accuracy relative to aggressive uniform co-quantization at similar or lower TTFT.

*Table 2.* End-to-end accuracy and speedup comparison on 8×RTX 4090. We evaluate on Arc-Challenge (AC), Arc-Easy (AE), HellaSwag (HS), LAMBADA-OpenAI (LO), LAMBADA-Standard (LS), PIQA (PQ), WinoGrande (WG), and WikiText-2 perplexity (PPL). WxAxCx denotes the bitwidth of weight-activation-communication. We use per-token × per-channel quantization by default except W4A4C4*, which using group size 128.

| Model | Method | AC↑ | AE↑ | HS↑ | LO↑ | LS↑ | PQ↑ | WG↑ | AVG↑ | PPL↓ | Speedup |
|---|---|---|---|---|---|---|---|---|---|---|---|
| | Full Precision | 55.29 | 78.32 | 75.51 | 71.69 | 66.12 | 78.51 | 71.27 | 70.96 | 7.67 | 1.0× |
| | W8A8C16 | 55.72 | 77.82 | 75.81 | 71.08 | 65.48 | 78.35 | 70.64 | 70.70 | 7.71 | 1.14× |
| | W4A4C16 | 41.38 | 62.67 | 69.44 | 55.56 | 49.14 | 72.04 | 64.25 | 59.21 | 10.24 | 1.16× |
| Llama3-8B-Instruct | W8A8C8 | 55.20 | 78.28 | 75.59 | 71.18 | 65.59 | 78.13 | 71.35 | 70.76 | 7.72 | 1.74× |
| | W4A4C4* | 40.87 | 62.08 | 68.88 | 60.45 | 53.95 | 73.18 | 64.88 | 60.61 | 9.21 | 2.78× |
| | CoCoQuant [1] | 53.33 | 76.60 | 74.83 | 69.88 | 64.54 | 76.55 | 71.19 | 69.56 | 7.87 | 2.73× |
| | CoCoQuant [2] | 56.14 | 78.20 | 75.59 | 71.43 | 66.21 | 77.58 | 70.24 | 70.77 | 7.72 | 2.13× |
| | Full Precision | 61.26 | 83.08 | 82.68 | 67.24 | 58.32 | 82.05 | 73.56 | 72.60 | 6.88 | 1.0× |
| | W8A8C16 | 60.92 | 83.04 | 82.26 | 68.81 | 59.21 | 81.72 | 72.14 | 72.59 | 6.99 | 1.22× |
| | W4A4C16 | 51.71 | 74.45 | 74.86 | 58.14 | 46.46 | 75.63 | 64.01 | 63.61 | 8.78 | 1.29× |
| Qwen3-32B | W8A8C8 | 61.69 | 83.08 | 82.49 | 68.27 | 59.25 | 81.56 | 72.85 | 72.74 | 6.98 | 1.76× |
| | W4A4C4* | 56.23 | 78.20 | 79.00 | 64.80 | 55.95 | 78.24 | 67.09 | 68.50 | 7.76 | 2.87× |
| | CoCoQuant [1] | 59.90 | 82.07 | 82.24 | 66.68 | 58.10 | 81.83 | 71.82 | 71.81 | 6.92 | 2.48× |
| | CoCoQuant [2] | 60.84 | 82.95 | 82.50 | 67.07 | 58.22 | 81.99 | 72.69 | 72.32 | 6.88 | 1.96× |
| | Full Precision | 63.05 | 82.45 | 85.00 | 76.13 | 72.54 | 84.82 | 82.79 | 78.11 | 3.63 | 1.0× |
| | W8A8C16 | 63.23 | 83.25 | 85.06 | 75.61 | 71.90 | 84.66 | 83.19 | 78.66 | 3.72 | 1.23× |
| | W4A4C16 | 53.58 | 76.81 | 77.38 | 64.74 | 59.29 | 78.94 | 73.64 | 69.20 | 7.30 | 1.31× |
| Llama3.3-70B-Instruct | W8A8C8 | 62.63 | 83.33 | 85.15 | 76.05 | 71.73 | 84.66 | 83.19 | 78.11 | 3.72 | 1.82× |
| | W4A4C4* | 55.38 | 78.79 | 81.73 | 71.26 | 65.30 | 81.45 | 76.87 | 72.97 | 6.28 | 3.01× |
| | CoCoQuant [1] | 62.63 | 82.95 | 84.94 | 75.96 | 72.52 | 84.60 | 82.00 | 77.94 | 3.85 | 2.61× |
| | CoCoQuant [2] | 62.54 | 82.91 | 85.06 | 76.48 | 72.37 | 84.82 | 83.43 | 78.23 | 3.66 | 2.09× |
| | Full Precision | 57.17 | 80.47 | 81.56 | 76.19 | 71.78 | 82.48 | 73.24 | 74.70 | 7.76 | 1.0× |
| | W8A8C16 | 56.57 | 80.09 | 81.64 | 76.29 | 71.63 | 82.26 | 72.22 | 74.39 | 8.02 | 1.14× |
| | W4A4C16 | 34.81 | 52.23 | 61.79 | 39.65 | 34.48 | 69.91 | 53.99 | 49.55 | 457.96 | 1.22× |
| Llama4-Scout-109B | W8A8C8 | 56.48 | 80.22 | 81.49 | 76.48 | 71.76 | 82.59 | 72.85 | 74.55 | 8.03 | 1.67× |
| | W4A4C4* | 54.61 | 77.02 | 78.43 | 71.92 | 66.06 | 79.87 | 66.61 | 70.65 | 18.44 | 2.45× |
| | CoCoQuant [1] | 55.12 | 78.66 | 80.87 | 72.17 | 67.32 | 80.79 | 70.24 | 72.17 | 10.49 | 2.55× |
| | CoCoQuant [2] | 56.31 | 80.09 | 81.56 | 76.31 | 71.73 | 82.32 | 71.9 | 74.32 | 8.02 | 1.88× |
| | Full Precision | 63.05 | 82.87 | 85.17 | 77.16 | 44.03 | 83.46 | 74.74 | 72.93 | 4.04 | 1.0× |
| | W8A8C16 | 63.48 | 83.08 | 85.33 | 77.35 | 44.54 | 82.64 | 75.14 | 73.08 | 4.05 | 1.23× |
| | W4A4C16 | 55.8 | 74.92 | 78.41 | 65.57 | 57.97 | 75.08 | 68.03 | 67.97 | 6.40 | 1.25× |
| Qwen3-MoE-235B | W8A8C8 | 63.14 | 83.12 | 85.06 | 77.37 | 45.27 | 83.46 | 75.69 | 73.30 | 4.05 | 1.75× |
| | W4A4C4* | 55.63 | 76.60 | 82.60 | 74.07 | 53.29 | 79.27 | 72.22 | 70.53 | 5.31 | 2.01× |
| | CoCoQuant [1] | 63.23 | 81.82 | 84.81 | 76.36 | 46.07 | 82.26 | 75.22 | 72.82 | 4.18 | 2.92× |
| | CoCoQuant [2] | 63.91 | 82.37 | 85.10 | 76.54 | 43.57 | 83.41 | 75.37 | 72.89 | 4.06 | 2.04× |

**Decode-stage scope.** Finally, we include a decode-stage profiling summary in Appendix G. The current implementation is primarily optimized for prefill/TTFT, where communication payloads are large enough for communication quantization and fused precision-boundary execution to dominate. Decode-heavy serving has smaller per-step payloads and is more sensitive to launch and synchronization overhead, so we treat decode acceleration as outside the main claim of this work.

### 4.4. Ablation Study

To validate the effectiveness of individual components in CoCoQuant, we conduct ablation studies focusing on three key aspects: the impact of Precision-aligned Graph Rewriting (PGR), the benefits of hardware-aware mixed-precision optimization, and the micro-level efficiency of our fused kernels.

**Impact of Precision-Aligned Graph Rewriting.** We compare the end-to-end performance of CoCoQuant with and without PGR under two representative uniform quantization configurations: aggressive **W4A4C4** (group size 128) and a standard **W8A8C8**. As reported in Table 3, PGR yields substantial gains in both model quality and inference speed. This results validate structural precision alignment is universally beneficial across different bit-width budgets and emphasis the importance of co-optimizing computation-communication instead of view quantized collective communication as opaque black-box.

**Effect of Mixed-Precision Optimization.** We evaluate our ILP-based allocation against uniform quantization under strict **Iso-Latency** constraints. As shown in Table 4,

*Table 3.* Impact of Precision-aligned Graph Re-writing (PGR) on Llama3.3-70B (8×RTX-4090). PGR simultaneously improves accuracy (lower PPL) and latency (higher Speedup) by eliminating redundant casts.

| Method | W4A4C4 | | W8A8C8 | |
|---|---|---|---|---|
| | PPL ↓ | Speedup ↑ | PPL ↓ | Speedup ↑ |
| w/o PGR | 6.29 | 2.78× | 3.76 | 1.74× |
| **w/ PGR** | **4.87** | **3.52×** | **3.66** | **2.05×** |

CoCoQuant demonstrates decisive advantages, particularly under aggressive budgets. While uniform W4A4 causes catastrophic collapse on Llama3-8B-Instruct (46.63%), CoCoQuant leverages operator heterogeneity to recover accuracy to **73.62%** (a **+27%** gain). By aggressively compressing robust FFN layers to preserve sensitive Attention layers, CoCoQuant identifies superior Pareto-optimal points that rigid strategies miss.

*Table 4.* **Effect of Mixed-Precision Optimization on GSM8K Accuracy.** We compare rigid Uniform quantization against Co-CoQuant's hardware-aware mixed-precision strategy under strict **Iso-Latency** constraints. While Uniform quantization suffers catastrophic degradation at low bit-widths (e.g., W4A4C4 with group size 128), CoCoQuant intelligently allocates bits to sensitive layers, recovering up to **27%** accuracy.

| Configuration | Llama3-8B | | Llama3-8B-Instruct | |
|---|---|---|---|---|
| | Acc (%) | Δ | Acc (%) | Δ |
| FP16 Baseline | 50.11 | - | 76.19 | - |
| *Scenario A: Moderate Budget (≈ W8A8C8 Latency)* | | | | |
| Uniform W8A8C8 | 46.93 | -3.18 | 76.12 | -0.07 |
| **CoCoQuant (Mixed)** | **47.76** | **-2.35** | **76.22** | **+0.03** |
| *Scenario B: Aggressive Budget (≈ W4A4C4 Latency)* | | | | |
| Uniform W4A4C4 | 21.23 | -28.88 | 46.63 | -29.56 |
| **CoCoQuant (Mixed)** | **43.82** | **-6.29** | **73.62** | **-2.57** |

**Micro-benchmarking of Fused Kernels.** Finally, we evaluate the efficiency of the fused kernel templates (Dequant-Reduce-Norm-Add-Quant) that underpin our graph rewriting. We benchmark the execution time of the fused kernel and calculate the speedup over the triton kernel generated by `torch.compile` (Ansel et al., 2024). Table 5 demonstrates that kernel fusion provides substantial speedup (up to **51.7×**) over `torch.compile`. Complete benchmark results for all 144 quantization configurations are provided in Appendix C.

## 5. Conclusion

This work demonstrates that breaking the distributed bandwidth wall requires rethinking quantization as a system-level design problem rather than a local operator transformation. CoCoQuant structurally integrates computation and communication into a coherent low-precision execu-

*Table 5.* Fused kernel performance (H800, batch size is 32K and hidden state is 5120). PT=Per-Token.

| Input | Output | Lat. ($\mu$s) | BW (TB/s) | Speedup |
|---|---|---|---|---|
| I4 Asym G128 | I4 Sym G64 | 48 | 1.54 | 51.7× |
| I4 Asym G64 | I4 Sym G64 | 49 | 1.56 | 51.1× |
| I4 Asym G128 | I8 Sym G64 | 49 | 1.60 | 50.1× |
| I8 Sym G128 | I4 Sym G128 | 64 | 1.65 | 25.6× |
| I8 Sym PT | I8 Sym PT | 63 | 1.72 | 6.9× |
| F8 Sym PT | F8 Sym PT | 63 | 1.73 | 4.3× |

tion model, pushing the Pareto frontier for scalable and hardware-efficiency distributed inference.

**Limitations and Scope.** CoCoQuant is currently most beneficial for prefill or TTFT-dominated serving. Decode-heavy workloads have much smaller communication payloads and are often latency-bound, where the multi-step A2A+Reduce+AG decomposition may not outperform specialized low-latency AllReduce kernels. This limitation is compatible with prefill-decode disaggregation: CoCo-Quant can be deployed on prefill instances to reduce TTFT, while decode instances retain existing low-latency communication paths.

## Acknowledgements

The authors would like to thank the diligent anonymous reviewers for their constructive feedback. We also extend our gratitude to the fla-org open-source community for their invaluable contributions to the linear attention infrastructure. This work is supported by the Shanghai Municipal Science and Technology Major Project.

## Impact Statement

This paper presents work whose goal is to advance the field of machine learning. There are many potential societal consequences of our work, none of which we feel must be specifically highlighted here.

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

## A. Layer-wise Quantization Sensitivity Analysis

**Sensitivity Heterogeneity.** We visualize layer-module quantization sensitivity as shown in Figure 3. Each heatmap shows the error distribution across layers (y-axis) and modules (x-axis) under fixed quantization precision. We observe strong heterogeneity across both dimensions: (i) different modules exhibit highly uneven sensitivity, and (ii) sensitivity varies significantly across layers within the same module. This effect is amplified under lower-bit quantization (4-bit vs. 8-bit), indicating that uniform precision assignment is structurally suboptimal. These observations empirically support the necessity of fine-grained mixed-precision and selective quantization strategies in LLMs.

## B. Additional Experiments

We conduct additional experiments on various models and complex reasoning tasks as table6 and Table 7 shows. We also conducted accuracy experiments on round-to-nearst weights shown in Table 8

*Table 6.* End-to-end accuracy and speedup comparison on 8×RTX 4090. We evaluate on Arc-Challenge (AC), Arc-Easy (AE), HellaSwag (HS), LAMBADA-OpenAI (LO), LAMBADA-Standard (LS), PIQA (PQ), WinoGrande (WG), and WikiText-2 perplexity (PPL). WxAxCx denotes the bitwidth of weight-activation-communication. We use per-token x per-channel quantization by default except W4A4C4*, which using group size 128.

| Model | Method | AC↑ | AE↑ | HS↑ | LO↑ | LS↑ | PQ↑ | WG↑ | AVG↑ | PPL↓ | Speedup |
|---|---|---|---|---|---|---|---|---|---|---|---|
| | Full Precision | 54.01 | 77.53 | 79.2 | 75.84 | 68.93 | 80.63 | 73.09 | 72.75 | 5.72 | 1.0× |
| Llama3-8B | W8A8C16 | 53.33 | 76.64 | 79.18 | 75.32 | 68.48 | 80.03 | 73.64 | 72.37 | 5.80 | 1.13× |
| | W4A4C16 | 38.14 | 62.08 | 68.75 | 56.64 | 51.19 | 72.47 | 64.00 | 59.04 | 9.26 | 1.16× |
| | W8A8C8 | 54.01 | 76.43 | 79.52 | 75.12 | 68.62 | 80.47 | 73.8 | 72.57 | 5.79 | 1.74× |
| | W8A8C4 | 53.16 | 76.98 | 78.96 | 74.79 | 67.98 | 79.71 | 72.93 | 72.07 | 5.87 | 2.52× |
| | W4A4C4* | 42.24 | 68.14 | 71.11 | 63.89 | 56.24 | 74.16 | 68.75 | 63.50 | 7.70 | 2.78× |
| | CoCoQuant [1] | 52.22 | 76.39 | 78.61 | 75.26 | 69.05 | 79.54 | 73.24 | 72.04 | 5.97 | 2.73× |
| | CoCoQuant [2] | 53.33 | 77.06 | 79.13 | 75.12 | 68.56 | 79.98 | 73.48 | 72.38 | 5.78 | 2.13× |
| | CoCoQuant [3] | 53.50 | 77.15 | 79.15 | 75.76 | 68.70 | 80.52 | 73.09 | 72.55 | 5.73 | 1.91× |

*Table 7.* GSM8K evaluation on various models

| Method | Llama3-8B-Instruct | Llama3-8B | Qwen3-32B |
|---|---|---|---|
| Full Precision | 76.19 | 50.11 | 72.861 |
| W8A8C16 | 76.19 | 47.92 | 72.93 |
| W8A8C8 | 76.12 | 46.93 | 71.27 |
| W4A4C4* | 46.63 | 21.23 | 70.58 |
| CoCoQuant [1] | 73.62 | 43.82 | 65.05 |
| CoCoQuant [2] | 76.22 | 47.76 | 73.31 |

## C. Full Kernel Benchmark Results

Due to the vast combination space of mixed precision, we could not write kernel for each combination. Suppose we have data format: BF16, FP8, INT8, INT4; group size: G256, G128, G64, G32, G16 and symmetric/asymmetric option. Considering fuse RMSNorm/Add/Gate Then we have 2000+ configurations:

$$(3 \times 5 \times 2 \times 2 + (3 \times 5 \times 2)^2) \times 3 = 2880$$

However, a unified kernel is also infeasible as it suffers from runtime overhead. we use C++ template to implement this kernel template, and specialize the template when nesscary.

Table 9 presents the complete benchmark results for our fused DQ→Reduce→RMSNorm→Add→Q kernel across all supported quantization configurations. All measurements are on RTX 4090 with input size 32K tokens × 5120 hidden dimension.

**Notation:** PT = Per-Token, G64/G128 = Group size 64/128, S/A = Symmetric/Asymmetric.

*Table 8.* RTN weight end-to-end comparison on Arc-Challenge (AC), Arc-Easy (AE), Hel- laSwag (HS), LAMBADA-OpenAI (LO), LAMBADA-Standard (LS), PIQA (PQ), WinoGrande (WG) and WikiText-2 perplexity (PPL). We use per-token x per-channel quantization by default except W4A4C4*, which using group size 128.

| Model | Method | AC↑ | AE↑ | HS↑ | LO↑ | LS↑ | PQ↑ | WG↑ | PPL↓ |
|---|---|---|---|---|---|---|---|---|---|
| Llama3-8B-Instruct | W8A8C16 | 57 | 78.87 | 75.88 | 71.51 | 66.18 | 78.67 | 71.51 | 7.70 |
| | W8A8C8 | 55.29 | 79.12 | 75.62 | 71.3 | 65.94 | 78.84 | 71.19 | 7.71 |
| | W4A4C4 | 23.98 | 28.54 | 27.52 | 0.41 | 0.16 | 50.49 | 49.41 | 5321.25 |
| | W4A4C4* | 44.54 | 65.91 | 70.75 | 59.93 | 53.13 | 73.99 | 65.82 | 10.73 |
| | CoCoQuant [1] | 55.29 | 79.12 | 75.62 | 74.44 | 69.01 | 78.84 | 71.19 | 5.84 |
| | CoCoQuant [2] | 44.54 | 65.91 | 70.75 | 62.93 | 56.2 | 73.99 | 65.82 | 8.73 |
| Llama3-8B | W8A8C16 | 57 | 78.87 | 75.88 | 74.89 | 68.45 | 78.67 | 71.51 | 5.83 |
| | W8A8C8 | 55.29 | 79.12 | 75.62 | 74.44 | 69.01 | 78.84 | 71.19 | 5.84 |
| | W4A4C4 | 23.98 | 28.54 | 27.52 | 0.35 | 0.08 | 50.49 | 49.41 | 5609.09 |
| | W4A4C4* | 44.54 | 65.91 | 70.75 | 62.93 | 56.2 | 73.99 | 65.82 | 8.73 |
| | CoCoQuant [1] | 56.14 | 8.07 | 75.76 | 70.87 | 65.46 | 78.62 | 71.27 | 7.72 |
| | CoCoQuant [2] | 49.06 | 70.41 | 72.72 | 63.71 | 56.69 | 75.68 | 67.56 | 9.66 |
| Qwen3-32B | W8A8C16 | 61.35 | 82.49 | 82.47 | 68.33 | 58.9 | 81.45 | 72.22 | 7.05 |
| | W8A8C8 | 61.26 | 82.49 | 82.2 | 68.15 | 59.31 | 81.23 | 72.85 | 7.06 |
| | W4A4C4 | 27.56 | 24.2 | 26.38 | 0 | 0 | 50.16 | 51.3 | 5.11e+9 |
| | W4A4C4* | 53.75 | 74.45 | 72.13 | 49.41 | 49.56 | 74.97 | 64.8 | 12.65 |
| | CoCoQuant [1] | 61.52 | 83.29 | 82.41 | 68.27 | 59.48 | 80.96 | 72.45 | 7.00 |
| | CoCoQuant [2] | 56.14 | 76.18 | 72.39 | 53.93 | 54.24 | 76.33 | 64.88 | 15.68 |
| Qwen3-MoE-235B | W8A8C16 | 64.33 | 82.74 | 85.2 | 75.57 | 43.1 | 83.73 | 75.45 | 4.10 |
| | W8A8C8 | 64.93 | 82.41 | 85 | 74.42 | 44.07 | 83.3 | 74.74 | 4.17 |
| | W4A4C4 | 23.81 | 27.95 | 27.8 | 0.35 | 0.47 | 50.6 | 49.72 | 2.57e+4 |
| | W4A4C4* | 55.97 | 78.87 | 81.83 | 65.59 | 49.7 | 78.35 | 69.61 | 5.65 |
| | CoCoQuant [1] | 64.51 | 82.49 | 85.28 | 74.95 | 44.15 | 83.62 | 76.09 | 4.10 |
| | CoCoQuant [2] | 60.49 | 79.97 | 83.72 | 69.47 | 51.19 | 80.47 | 70.24 | 5.01 |
| Llama4-Scout-109B | W8A8C16 | 61.35 | 82.49 | 82.47 | 76.93 | 72.06 | 81.45 | 72.22 | 8.21 |
| | W8A8C8 | 61.26 | 82.49 | 82.2 | 75.53 | 68.56 | 81.23 | 72.85 | 9.10 |
| | W4A4C4 | 27.56 | 24.2 | 26.38 | 0 | 0 | 50.16 | 51.3 | 7.31e+19 |
| | W4A4C4* | 53.75 | 74.45 | 72.13 | 67.44 | 63.52 | 74.97 | 64.8 | 23.49 |
| | CoCoQuant [1] | 61.52 | 83.29 | 82.41 | 75.51 | 68.68 | 80.96 | 72.45 | 9.05 |
| | CoCoQuant [2] | 56.14 | 76.18 | 72.39 | 69.61 | 65.2 | 76.33 | 64.88 | 21.79 |

*Table 9.* Complete fused kernel benchmark results across all input/output configurations.

| Input | Output | Fused (ms) | Unfused (ms) | Speedup |
|---|---|---|---|---|
| *FP8 Symmetric Per-Token Input* | | | | |
| FP8 S PT | FP8 S PT | 0.063 | 0.268 | 4.26× |
| FP8 S PT | INT8 S PT | 0.063 | 0.432 | 6.81× |
| FP8 S PT | INT8 A PT | 0.064 | 0.338 | 5.32× |
| FP8 S PT | FP8 S G64 | 0.057 | 0.375 | 6.54× |
| FP8 S PT | INT8 S G64 | 0.058 | 1.850 | 32.15× |
| FP8 S PT | INT8 A G64 | 0.057 | 1.683 | 29.33× |
| FP8 S PT | FP8 S G128 | 0.063 | 0.335 | 5.34× |
| FP8 S PT | INT8 S G128 | 0.063 | 1.231 | 19.54× |
| FP8 S PT | INT8 A G128 | 0.063 | 1.064 | 16.81× |
| FP8 S PT | INT4 S PT | 0.063 | 0.463 | 7.32× |
| FP8 S PT | INT4 A PT | 0.064 | 0.367 | 5.69× |
| FP8 S PT | INT4 S G64 | 0.057 | 1.878 | 33.01× |
| FP8 S PT | INT4 A G64 | 0.057 | 1.711 | 30.08× |
| FP8 S PT | INT4 S G128 | 0.063 | 1.260 | 20.17× |
| FP8 S PT | INT4 A G128 | 0.063 | 1.092 | 17.46× |
| *INT8 Symmetric Per-Token Input* | | | | |
| INT8 S PT | FP8 S PT | 0.063 | 0.271 | 4.28× |
| INT8 S PT | INT8 S PT | 0.063 | 0.435 | 6.87× |
| INT8 S PT | INT8 A PT | 0.063 | 0.340 | 5.36× |
| INT8 S PT | FP8 S G64 | 0.058 | 0.376 | 6.47× |
| INT8 S PT | INT8 S G64 | 0.059 | 1.853 | 31.54× |
| INT8 S PT | INT8 A G64 | 0.058 | 1.685 | 29.03× |
| INT8 S PT | FP8 S G128 | 0.063 | 0.337 | 5.39× |
| INT8 S PT | INT8 S G128 | 0.063 | 1.232 | 19.56× |
| INT8 S PT | INT8 A G128 | 0.063 | 1.066 | 16.99× |
| INT8 S PT | INT4 S PT | 0.063 | 0.464 | 7.34× |
| INT8 S PT | INT4 A PT | 0.065 | 0.369 | 5.72× |
| INT8 S PT | INT4 S G64 | 0.057 | 1.880 | 32.88× |
| INT8 S PT | INT4 A G64 | 0.057 | 1.714 | 29.87× |
| INT8 S PT | INT4 S G128 | 0.063 | 1.261 | 20.19× |
| INT8 S PT | INT4 A G128 | 0.063 | 1.094 | 17.49× |
| *INT8 Asymmetric Per-Token Input* | | | | |
| INT8 A PT | FP8 S PT | 0.064 | 0.304 | 4.78× |
| INT8 A PT | INT8 S PT | 0.064 | 0.468 | 7.33× |
| INT8 A PT | INT8 A PT | 0.064 | 0.373 | 5.85× |
| INT8 A PT | FP8 S G64 | 0.059 | 0.410 | 7.00× |
| INT8 A PT | INT8 S G64 | 0.059 | 1.885 | 31.91× |
| INT8 A PT | INT8 A G64 | 0.059 | 1.717 | 29.32× |
| INT8 A PT | FP8 S G128 | 0.063 | 0.370 | 5.85× |
| INT8 A PT | INT8 S G128 | 0.063 | 1.266 | 20.11× |
| INT8 A PT | INT8 A G128 | 0.063 | 1.098 | 17.39× |
| INT8 A PT | INT4 S PT | 0.064 | 0.497 | 7.75× |
| INT8 A PT | INT4 A PT | 0.065 | 0.402 | 6.24× |
| INT8 A PT | INT4 S G64 | 0.058 | 1.914 | 33.00× |
| INT8 A PT | INT4 A G64 | 0.058 | 1.745 | 30.05× |
| INT8 A PT | INT4 S G128 | 0.063 | 1.294 | 20.72× |
| INT8 A PT | INT4 A G128 | 0.063 | 1.127 | 17.81× |
| *FP8/INT8 Group-64 Input* | | | | |
| FP8 S G64 | FP8 S PT | 0.064 | 0.652 | 10.18× |
| FP8 S G64 | INT8 S G64 | 0.063 | 2.234 | 35.40× |
| FP8 S G64 | INT4 S G64 | 0.062 | 2.261 | 36.69× |
| INT8 S G64 | FP8 S PT | 0.064 | 0.650 | 10.12× |
| INT8 S G64 | INT8 S G64 | 0.063 | 2.232 | 35.38× |
| INT8 S G64 | INT4 S G64 | 0.062 | 2.261 | 36.42× |
| INT8 A G64 | INT8 S G64 | 0.061 | 2.265 | 37.35× |

*(continued on next page)*

*(continued from previous page)*

| Input | Output | Fused (ms) | Unfused (ms) | Speedup |
|---|---|---|---|---|
| INT8 A G64 | INT4 S G64 | 0.059 | 2.294 | 38.66× |
| *FP8/INT8 Group-128 Input* | | | | |
| FP8 S G128 | FP8 S PT | 0.063 | 0.653 | 10.30× |
| FP8 S G128 | INT8 S G64 | 0.062 | 2.235 | 36.35× |
| FP8 S G128 | INT4 S G64 | 0.060 | 2.261 | 37.59× |
| INT8 S G128 | FP8 S PT | 0.064 | 0.651 | 10.18× |
| INT8 S G128 | INT8 S G64 | 0.062 | 2.232 | 35.90× |
| INT8 S G128 | INT4 S G64 | 0.061 | 2.261 | 37.07× |
| INT8 A G128 | INT8 S G64 | 0.060 | 2.266 | 37.76× |
| INT8 A G128 | INT4 S G64 | 0.059 | 2.294 | 39.00× |
| *INT4 Symmetric Per-Token Input* | | | | |
| INT4 S PT | FP8 S PT | 0.063 | 0.551 | 8.73× |
| INT4 S PT | INT8 S PT | 0.064 | 0.716 | 11.19× |
| INT4 S PT | INT8 A PT | 0.063 | 0.619 | 9.79× |
| INT4 S PT | FP8 S G64 | 0.053 | 0.657 | 12.38× |
| INT4 S PT | INT8 S G64 | 0.054 | 2.131 | 39.59× |
| INT4 S PT | INT8 A G64 | 0.053 | 1.966 | 36.97× |
| INT4 S PT | FP8 S G128 | 0.055 | 0.617 | 11.16× |
| INT4 S PT | INT8 S G128 | 0.056 | 1.514 | 27.21× |
| INT4 S PT | INT8 A G128 | 0.055 | 1.345 | 24.46× |
| INT4 S PT | INT4 S PT | 0.063 | 0.743 | 11.75× |
| INT4 S PT | INT4 A PT | 0.064 | 0.648 | 10.21× |
| INT4 S PT | INT4 S G64 | 0.054 | 2.159 | 40.12× |
| INT4 S PT | INT4 A G64 | 0.054 | 1.994 | 36.73× |
| INT4 S PT | INT4 S G128 | 0.056 | 1.542 | 27.56× |
| INT4 S PT | INT4 A G128 | 0.056 | 1.375 | 24.58× |
| *INT4 Asymmetric Per-Token Input* | | | | |
| INT4 A PT | FP8 S PT | 0.059 | 0.482 | 8.17× |
| INT4 A PT | INT8 S PT | 0.059 | 0.649 | 11.09× |
| INT4 A PT | INT8 A PT | 0.059 | 0.553 | 9.42× |
| INT4 A PT | FP8 S G64 | 0.047 | 0.589 | 12.44× |
| INT4 A PT | INT8 S G64 | 0.048 | 2.064 | 43.42× |
| INT4 A PT | INT8 A G64 | 0.048 | 1.897 | 39.74× |
| INT4 A PT | FP8 S G128 | 0.048 | 0.549 | 11.35× |
| INT4 A PT | INT8 S G128 | 0.048 | 1.446 | 29.89× |
| INT4 A PT | INT8 A G128 | 0.049 | 1.278 | 26.35× |
| INT4 A PT | INT4 S PT | 0.059 | 0.676 | 11.48× |
| INT4 A PT | INT4 A PT | 0.060 | 0.582 | 9.77× |
| INT4 A PT | INT4 S G64 | 0.047 | 2.092 | 44.50× |
| INT4 A PT | INT4 A G64 | 0.048 | 1.925 | 40.52× |
| INT4 A PT | INT4 S G128 | 0.048 | 1.473 | 30.49× |
| INT4 A PT | INT4 A G128 | 0.049 | 1.307 | 26.79× |
| *INT4 Symmetric Group-64 Input* | | | | |
| INT4 S G64 | FP8 S PT | 0.064 | 0.930 | 14.42× |
| INT4 S G64 | INT8 S PT | 0.066 | 1.094 | 16.66× |
| INT4 S G64 | INT8 S G64 | 0.059 | 2.510 | 42.28× |
| INT4 S G64 | INT8 A G64 | 0.054 | 2.343 | 43.43× |
| INT4 S G64 | INT4 S G64 | 0.059 | 2.540 | 42.93× |
| INT4 S G64 | INT4 A G64 | 0.055 | 2.372 | 42.78× |
| *INT4 Asymmetric Group-64 Input* | | | | |
| INT4 A G64 | FP8 S PT | 0.061 | 0.866 | 14.12× |
| INT4 A G64 | INT8 S PT | 0.061 | 1.030 | 16.81× |
| INT4 A G64 | FP8 S G64 | 0.049 | 0.977 | 19.98× |
| INT4 A G64 | INT8 S G64 | 0.049 | 2.451 | **49.79×** |
| INT4 A G64 | INT8 A G64 | 0.049 | 2.285 | 46.29× |
| INT4 A G64 | INT4 S G64 | 0.049 | 2.482 | **51.07×** |

*(continued from previous page)*

| Input | Output | Fused (ms) | Unfused (ms) | Speedup |
|---|---|---|---|---|
| INT4 A G64 | INT4 A G64 | 0.050 | 2.312 | 46.66× |
| *INT4 Symmetric Group-128 Input* | | | | |
| INT4 S G128 | FP8 S PT | 0.065 | 0.930 | 14.39× |
| INT4 S G128 | INT8 S PT | 0.065 | 1.093 | 16.79× |
| INT4 S G128 | INT8 S G64 | 0.059 | 2.510 | 42.75× |
| INT4 S G128 | INT8 A G64 | 0.054 | 2.343 | 43.56× |
| INT4 S G128 | INT4 S G64 | 0.059 | 2.538 | 43.19× |
| INT4 S G128 | INT4 A G64 | 0.055 | 2.371 | 43.19× |
| *INT4 Asymmetric Group-128 Input* | | | | |
| INT4 A G128 | FP8 S PT | 0.060 | 0.866 | 14.40× |
| INT4 A G128 | INT8 S PT | 0.060 | 1.030 | 17.25× |
| INT4 A G128 | FP8 S G64 | 0.048 | 0.972 | 20.05× |
| INT4 A G128 | INT8 S G64 | 0.049 | 2.446 | **50.11×** |
| INT4 A G128 | INT8 A G64 | 0.049 | 2.278 | 46.79× |
| INT4 A G128 | INT4 S G64 | 0.048 | 2.475 | **51.66×** |
| INT4 A G128 | INT4 A G64 | 0.049 | 2.308 | 47.02× |
| INT4 A G128 | INT4 S G128 | 0.049 | 1.856 | 37.85× |
| INT4 A G128 | INT4 A G128 | 0.050 | 1.688 | 33.52× |

**Observations:** The fused kernel achieves consistent latency (∼0.05–0.07ms) regardless of input/output configurations, while the `torch.compile` baseline varies significantly (0.27–2.54ms). Maximum speedup of **51.66×** is achieved with INT4 Asymmetric Group-128 input to INT4 Symmetric Group-64 output. Group quantization outputs consistently achieve higher speedups (30–50×) than per-token outputs (4–17×) due to the uncovered pattern in `torch.compile`, which failed to fuse all the operation into one single kernel and fallback to torch implementation.

## D. Extra Datacenter GPU Evaluation

Figure 4 analyzes how CoCoQuant performance on datacenter GPU H800 with NVLink(capped, 200GB/s) under a fixed total token budget of 32K tokens. As batch size increases (with correspondingly shorter sequence lengths), we observe:

**(1) Consistent speedup across configurations:** CoCoQuant achieves 1.5–1.8× speedup on TP4 and 1.4–1.6× on TP8 across all batch sizes, demonstrating robust performance regardless of the batch/sequence/TP-size trade-off.

**(2) Higher speedup at larger batch sizes:** Within each TP configuration, speedup slightly increases with batch size (e.g., Llama3.3-70B on TP4: 1.67× at batch=2 → 1.79× at batch=16). This is because larger batches with shorter sequences reduce the relative contribution of attention computation (which scales quadratically with sequence length), making the communication overhead—where our compression provides the most benefit—more prominent.

**(3) Negligible accuracy impact:** We set latency budget to 60%, so the results is expected lossless. As shown in the figure titles, the PPL degradation is minimal (<0.01) across all models, confirming that our mixed-precision allocation preserves model quality while delivering substantial speedups.

## E. Fine-Grained Prefill Breakdown

This section expands the brief system discussion in subsection 4.3. We profile the same vLLM-based tensor-parallel stack used in the main evaluation and decompose prefill latency into attention kernels, quantized GEMMs, communication, CoCoQuant fused boundary kernels, and an "Other" category. The "Other" category captures residual row-wise kernels, Q/DQ transitions, launch overhead, and runtime work not included in the first four groups. This breakdown separates the two questions raised by the end-to-end numbers: whether CoCoQuant reduces communication/computation work, and whether the gain depends on comparing against an unfused baseline.

All rows use Qwen3-32B with TP8 and batch size 4 on GovReport from LongBench, whose prompts average around 8K tokens. The `vLLM SP W8A8C8` row is the strongest uniform communication-quantized SP baseline in this comparison and already uses the A2A+Reduce+AG communication pattern. The `CoCoQuant w/o Fused` row keeps the same

*Table 10.* Absolute prefill latency breakdown for the same runs as Figure 5. Entries are milliseconds with share of traced GPU activity in parentheses.

| Method | Pattern | Prefill | Attn | GEMM | Comm. | CQ-Fused | Other |
|--------|---------|---------|------|------|-------|----------|-------|
| vLLM TP BF16 | AR | 7062.02 | 256.53 (3.6%) | 1488.97 (21.1%) | 4947.79 (70.1%) | 0.00 (0.0%) | 365.84 (5.2%) |
| vLLM SP BF16 | RS+AG | 7373.55 | 256.05 (3.5%) | 1487.11 (20.2%) | 5534.66 (75.1%) | 0.00 (0.0%) | 90.44 (1.2%) |
| vLLM SP W8A8 | RS+AG | 6359.86 | 255.69 (4.0%) | 1044.67 (16.4%) | 5014.01 (78.9%) | 0.00 (0.0%) | 41.95 (0.7%) |
| vLLM SP W8A8C8 | A2A+Reduce+AG | 3768.26 | 255.30 (6.8%) | 745.40 (19.8%) | 2570.51 (68.3%) | 168.51 (4.5%) | 24.87 (0.7%) |
| CoCoQuant | A2A+Reduce+AG | **2889.59** | **254.33 (8.8%)** | **699.60 (24.2%)** | **1755.03 (60.8%)** | **152.01 (5.3%)** | **24.69 (0.9%)** |
| CoCoQuant w/o Fused | A2A+Reduce+AG | 3797.74 | 254.33 (6.7%) | 699.60 (18.4%) | 1755.03 (46.2%) | 0.00 (0.0%) | 1088.77 (28.7%) |

CoCoQuant bit choices and communication pattern but disables the precision-aligned fused path, isolating the contribution of fusion and boundary elimination.

## F. Layer-wise Bit Allocation Patterns

To inspect how the ILP uses the expanded communication-computation search space, we visualize the layer-wise bit allocation in Figure 6. The figure compares three points: the uniform `W8A8C4` baseline, a matched-latency CoCoQuant point (CoCoQuant *), and the CoCoQuant configuration used in the main experiments (CoCoQuant[2]). Each row group corresponds to a model component or communication boundary, and each column corresponds to a layer.

The allocation is structured rather than uniform. The matched-latency CoCoQuant * configuration spends its precision budget on modules where uniform low-bit communication causes the largest perplexity degradation, while using lower precision on robust middle-layer blocks to preserve latency. The main table configuration is more accuracy-oriented and keeps additional sensitive boundaries at higher precision. This supports the role of the ILP as a hardware-aware selection mechanism: it converts the latency budget into layer-specific precision choices instead of applying a fixed global communication bit-width.

## G. Decode-Scope Analysis

*Table 11.* Decode-stage profiling after the prefill span. The first-step kernel-only time highlights that decode-heavy serving is latency-bound and may favor specialized low-latency communication kernels.

| Method | First Post Event | First Step Kernel-Only | Comm. | GEMM |
|--------|------------------|------------------------|-------|------|
| vLLM TP BF16 | 19.52 | 19.28 | 24.93 (32.1%) | 41.23 (53.2%) |
| vLLM SP W8A8C8 | 37.02 | 33.54 | 239.56 (84.4%) | 24.02 (8.5%) |
| CoCoQuant-20 | 41.55 | 26.74 | 242.70 (84.3%) | 25.84 (9.0%) |
| CoCoQuant-30 | 37.43 | 29.08 | 219.91 (83.0%) | 24.85 (9.4%) |

Unlike prefill, decode communicates small per-step payloads and is more sensitive to launch and synchronization latency. Therefore, CoCoQuant should be viewed primarily as a TTFT/prefill optimization in the current implementation, while decode-heavy deployments can retain specialized low-latency communication paths.

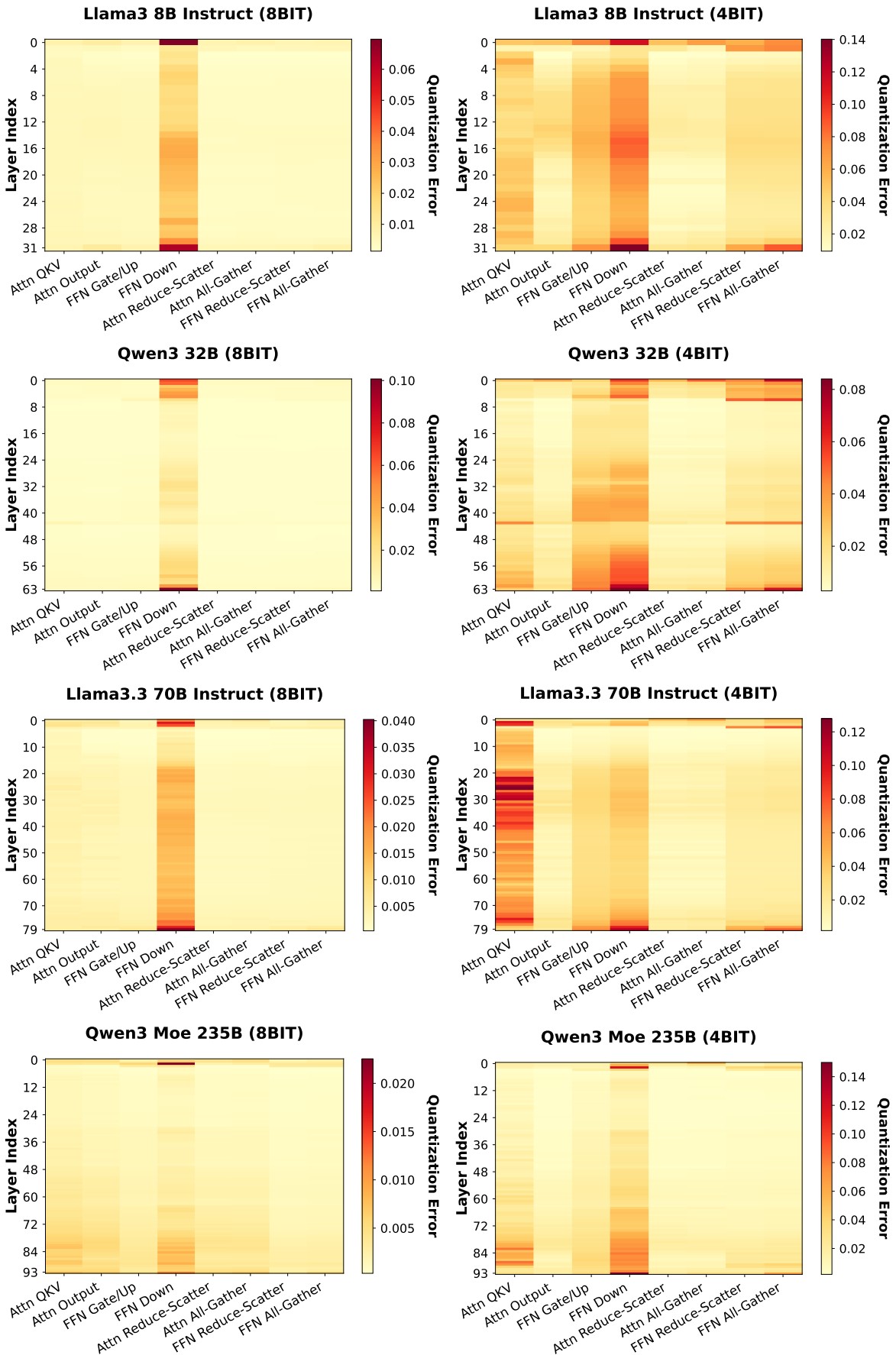

*Figure 3.* Layer-module quantization sensitivity heatmaps. For each model: left is 8-bit, right is 4-bit. Color indicates quantization error.

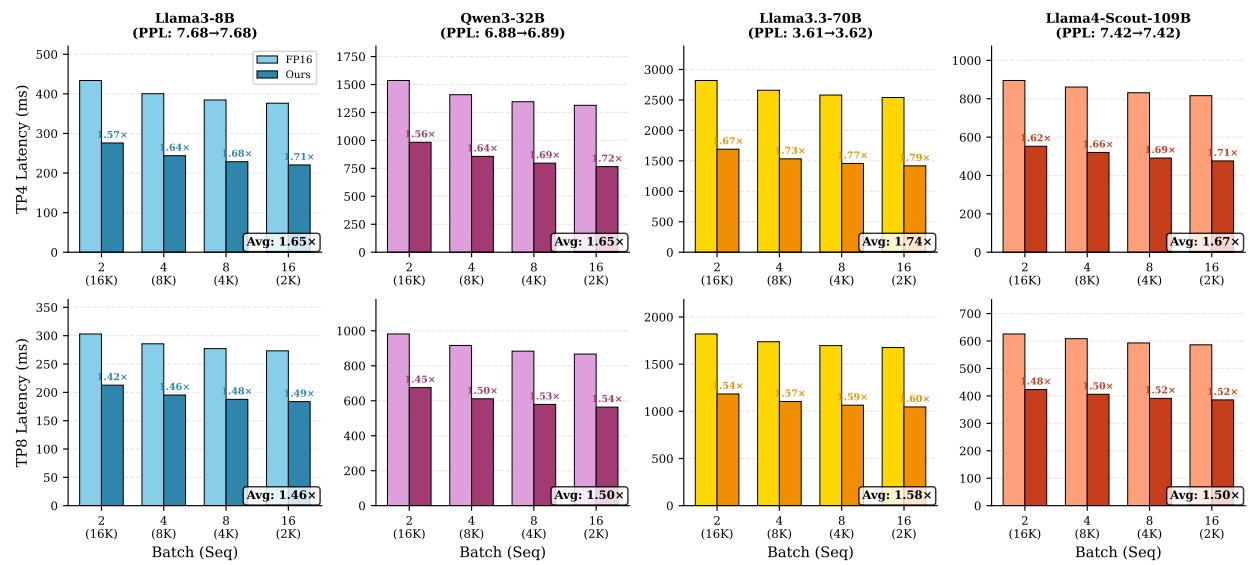

*Figure 4.* Latency and speedup across different batch sizes under fixed total token count (32K tokens). Top row: TP4 (4 GPUs); Bottom row: TP8 (8 GPUs). Each bar pair shows FP16 baseline (light) vs. CoCoQuant (dark), with speedup annotated above our bars.

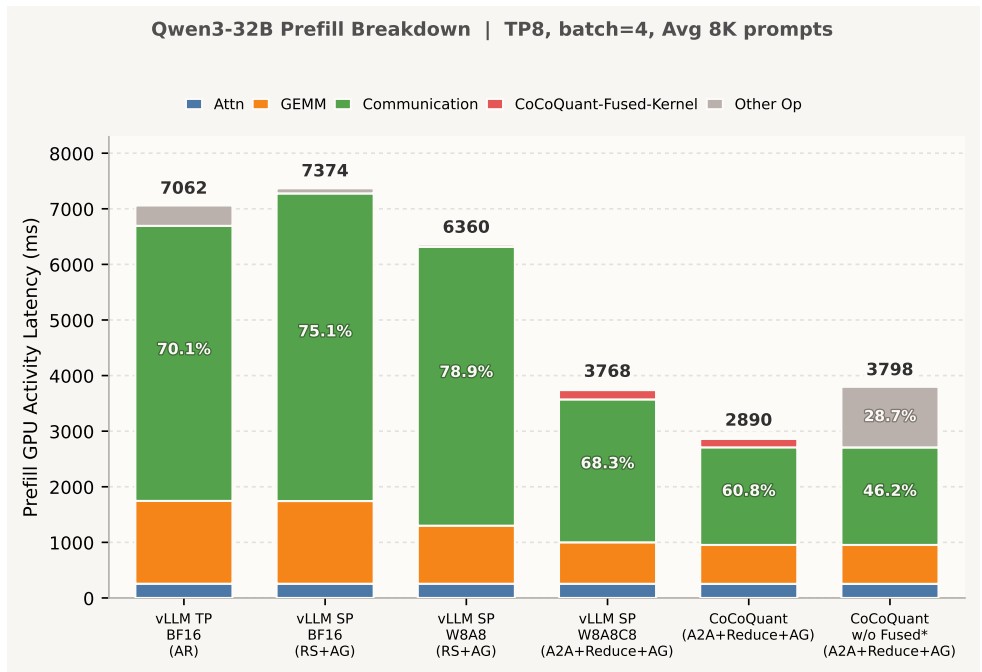

*Figure 5.* Prefill latency breakdown on Qwen3-32B with TP8 and batch size 4, evaluated on GovReport from LongBench with average prompt length around 8K tokens. Numbers inside bars denote communication share.

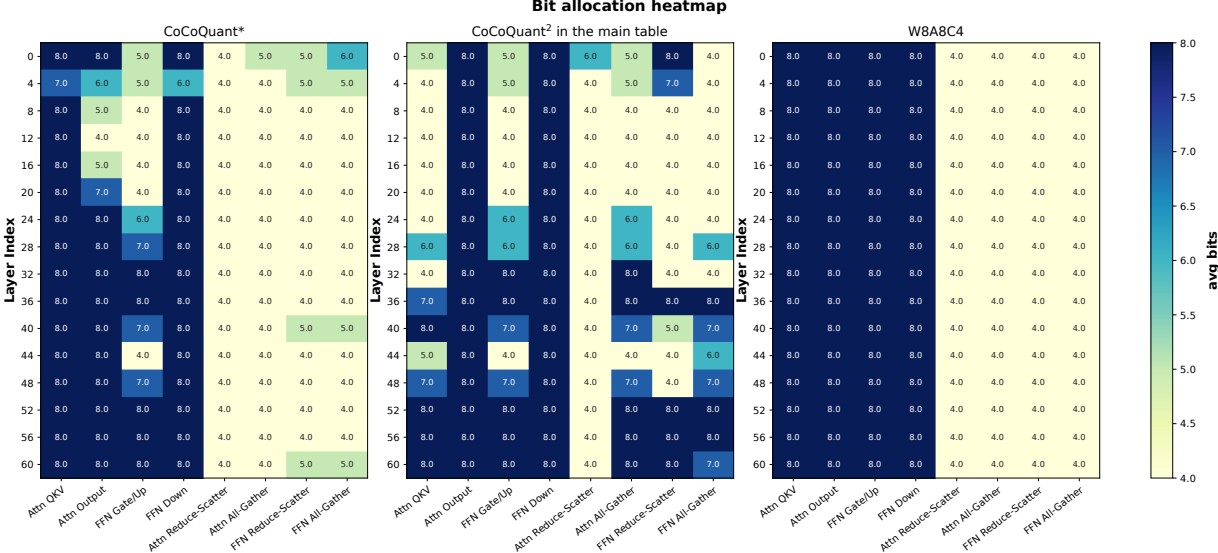

*Figure 6.* Layer-wise bit-allocation heatmap comparing the uniform `W8A8C4` baseline, a matched-latency CoCoQuant point, and the CoCoQuant configuration used in the main table. CoCoQuant protects sensitive early/late communication boundaries and selected FFN modules while assigning lower precision to more robust middle-layer modules.

