# OpenReview forum: "CoCoQuant: Breaking the Bandwidth Wall via Co-Optimized Communication and Computation Quantization"
_ICML.cc/2026/Conference — ICML 2026 regular_

### Official Review · Reviewer_ynLp · 2026-03-10

**Soundness:** 3
**Presentation:** 3
**Significance:** 2
**Originality:** 2
**Overall Recommendation:** 4
**Confidence:** 3

**Summary:**

This paper proposes CoCoQuant, a framework that formulates communication and computation quantization as a unified optimization space rather than treating them separately. The approach jointly optimizes communication and computation precision to reduce redundant quantization/dequantization overhead, and uses an ILP-based mixed-precision allocation method to select precisions for different operators. The authors report up to 2.92× speedup with modest perplexity degradation across several dense and MoE models.

**Compliance With Llm Reviewing Policy:**

Affirmed.

**Final Justification:**

The rebuttal addressed my main concerns. In particular, the new ablation study is informative and clearly demonstrates the source of the speedup.

**Key Questions For Authors:**

1. Could the authors provide a latency breakdown of the serving pipeline, including the latency of individual operations? This would help clarify where the performance gains come from and which components contribute most to the overall savings.
2. In addition to fixed precision allocation methods, could the authors compare against prior adaptive or mixed-precision approaches?
3. One claimed contribution is moving element-wise and row-wise operations from the replicated phase to the sharded phase. However, my understanding is that these operations are not always in the replicated phase by default. For example, in NVIDIA’s sequence parallelism implementation, dropout, residual addition, and LayerNorm are placed after reduce-scatter and before all-gather  [1].
4. Could the authors provide more details about the baseline implementations? In particular, is the baseline a completely unfused implementation? In modern frameworks such as Megatron and CUTLASS, epilogue fusion is already quite common.
5. The primary gains of the proposed method appear to come from graph rewriting, kernel fusion, and mixed-precision allocation. Since these techniques are already broadly used, could the authors clarify what is fundamentally new or novel in their specific combination or application?

**Reference:**

[1] Korthikanti, Vijay Anand, et al. "Reducing activation recomputation in large transformer models." Proceedings of Machine Learning and Systems 5 (2023): 341-353.

**Limitations:**

Yes

**Strengths And Weaknesses:**

**Strength:**
* The method is well motivated by the communication bottleneck in distributed LLM inference.
* Jointly optimizing communication and computation precision is an interesting idea.
* The empirical results are promising.

**Weakness:**
* The overall contribution appears somewhat incremental. The paper combines graph rewriting, mixed-precision allocation, and kernel fusion, all of which have been well explored in prior work. It remains unclear what is fundamentally new beyond integrating these existing ideas.
* The experimental comparison is incomplete. The evaluation mainly compares against fixed-precision baselines, but a stronger comparison would include prior adaptive or mixed-precision allocation methods.
* The baseline implementation is unclear. Since the reported gains depend on graph rewriting and kernel fusion, the paper should more clearly specify the baseline configuration and implementation details, including whether it already incorporates standard modern optimizations such as epilogue fusion.

---

> ### Author Rebuttal · Authors · 2026-03-30
>
> We thank the reviewer for the constructive feedback. To clarify our contributions within space limits, we provide detailed experiment results in this anonymous link: https://1drv.ms/b/c/316d36085eac4395/IQDWtM23weHmT57t2AprhCsZAeBMxYC0yQN8AhHX1Hj-zhM?e=d4mkOb. We address your questions below:
>
> > ## **1. Q1: Latency Breakdown & Comparison with Baselines (CR4 - Common Response 4)**
>
> Please refer to attached anonymous PDF (Section A) for the detailed result table. we also provide a concise copy here:
> |Method|Pattern|TTFT (ms)|Attn (ms)|GEMM (ms)|Comm (ms)|CQ-Fused (ms)|Other (ms)|
> |-|-:|-:|-:|-:|-:|-:|-:|
> |vLLM TP BF16|AR|7062.0|256.5|1489.0|4947.8|0.0|365.8|
> |vLLM SP BF16|RS+AG|7373.6|256.1|1487.1|5534.7|0.0|90.4|
> |**vLLM SP W8A8**|RS+AG|**6359.9**|255.7|1044.7|5014.0|0.0|42.0|
> |vLLM SP W8A8C8|A2A+Red+AG|3768.3|255.3|745.4|2570.5|168.5|24.9|
> |**CoCoQuant**|A2A+Red+AG|**2889.6**|254.3|699.6|1755.0|152.0|**24.7**|
> |CoCoQuant w/o Fused| A2A+Red+AG|3797.7|254.3|699.6|1755.0|0.0|**1088.8**|
>
> 1. **Compute-only quantization hits a bandwidth wall:** In optimized `vLLM SP W8A8`, communication still consumes **78.9% (5014.0 ms)** of TTFT.
> 2. **ILP outperforms uniform co-quantization:** Our ILP policy (matching the `CoCoQuant`$^1$ in Table 2) pushes TTFT to **2889.6 ms**—a **1.3x speedup** over uniform W8A8C8, ensuring a superior Pareto frontier.
> 3. **Graph Rewriting & Fusion are essential:** Without our custom kernels (`CoCoQuant w/o Fused`), Q/DQ casting overhead ("Other") explodes from 24.7 ms to **1088.8 ms (28.7% of latency)**. Precision fragmentation completely wipes out co-quantization benefits without our co-design.
>
> > ## **2. W1, Q3, Q5: Incremental Contribution & Relation to Sequence Parallelism**
>
> While moving operators resembles Megatron's SP, and ILP/fusions are known, we respectfully argue that **the commonality of individual techniques does not diminish the novelty of orchestrating them to unblock a new paradigm**. FlashAttention, for instance, revolutionized Transformers by repurposing decades-old HPC tiling and fusion to solve a specific memory bottleneck. Similarly, CoCoQuant's fundamental novelty is an Algorithm-System Co-design solving a new problem:
> 1. **A Holistic Co-Quantization Paradigm:** Comm and comp compression historically treated as isolated silos; optimizing one shifts the bottleneck to the other. CoCoQuant is the first to jointly optimize both, incorporating hardware latency for Pareto optimality.
> 2. **Identifying the "Precision Fragmentation" Problem:** We made a critical unprecedented observation: **Unconstrained bit-width allocation maximizes theoretical accuracy but physically introduces severe Q/DQ overheads at comm-comp boundaries, wiping out system acceleration.** We thus explicitly co-design the algorithm by adding alignment constraints (Eqs. 4-5), restricting the algorithmic search space to guarantee executable hardware acceleration.
> 3. **Unprecedented Solutions:** To resolve this fragmentation, our proposed Precision-Aligned Graph Rewriting (PGR) and cross-boundary kernels are entirely unprecedented. (For PGR's divergence from standard SP, due to space limitation, please refer to our **response to Reviewer WYpS (Common Response 1, CR1)**). Our breakdown explicitly validates that without this co-design, co-quantization remains a theoretical exercise.
>
> CoCoQuant is the first work to explore the co-quantization and formulate the precision alignment problem. We employ these techniques because they practically solve the bottleneck we identified. This specific combination is not a simple engineering patch, but an indispensable co-design paradigm.
> > ## **3. W3, Q4: Baseline Implementation and Fusion Details**
>
> Due to space limitations, please refer to **our Responses to Reviewer dFpG (CR2)** for more details about the baseline implementation.
>
> Our cross-collective fused kernels are fundamentally different from standard GEMM + element-wise epilogue fusions in CUTLASS, because existing generic fusions (and compilers) cannot natively cross into quantized distributed communication boundaries. Seamlessly fusing `DQ->Reduce->MoE Gate->Q` is the indispensable technical step that translates graph rewriting into actual latency reductions.
> > ## **4. W2, Q2: Comparison Against Mixed-Precision Baselines**
>
> Existing mixed-precision methods (e.g., HAWQ) are inherently compute-centric; they do not jointly optimize communication. To address your concern, we implement a Hessian-based Mixed-Precision Quantization (HMQ) widely used in mixed-precision methods like HAWQ and HAQ. HMQ adaptively allocates compute bit-widths using second-order info metric but leaves communication isolated (settings match Table 2, test on 8xH800).
>
> **Please refer to the attached anonymous PDF for the detailed result table:** With virtually identical accuracy, CoCoQuant delivers **17.1%–27.1%** e2e latency reductions over existing mixed-precision methods. This proves our joint search space strictly expands the Pareto frontier.

---

> > ### Author Rebuttal · Reviewer_ynLp · 2026-04-03
> >
> > Thank you for the response.
> >
> > I noticed that CoCoQuant achieves lower latency in GEMM and communication compared to the vLLM SP W8A8C8 baseline. Could you elaborate more on why it achieves lower latency? And how do the latency and accuracy compare to vLLM SP W8A8C4?

---

> > > ### Author Response · Authors · 2026-04-03
> > >
> > > Thank you for the insightful follow-up questions.
> > >
> > > > ## 1. **Why CoCoQuant achieves lower latency than uniform W8A8C8:**
> > >
> > > As defined by our objective in Eq. 11, CoCoQuant leverages an ILP solver to identify robust computation and communication modules, safely pushing them to lower bit-widths (e.g., 4-bit). While the uniform W8A8C8 baseline is rigidly constrained to 8-bit across all modules, CoCoQuant aggressively optimizes insensitive layers to drive overall latency down.
> > >
> > > To concretely illustrate where these latency savings come from, we extract the actual bit allocations for early (0), middle (44), and late (63) layers, compared to the new uniform W8A8C4 baseline:
> > >
> > > - **NOTE:** we also provide a more detailed and easy-to-read figure in the **new anoymous link: https://1drv.ms/b/c/316d36085eac4395/IQDajipVWHucS7PigAiBPCj-AWUy6NGF3y5GEbi9uwIcgaY?e=4uNob9**
> > >
> > > | Layer | Module | W8A8C4 | CoCoQuant$^*$ | CoCoQuant |
> > > |:---:|---|---|---|---|
> > > | **Layer 0** | Attn QKV / Output | INT8 | INT8 | INT8 |
> > > | | FFN Down | INT8 | **FP8** | **FP8** |
> > > | | Attn / FFN Comm | **INT4** | INT8 / INT4 | **INT8** |
> > > | **Layer 44** | Attn / FFN Comm | INT4 | **INT4** | INT4 / INT8 |
> > > | | FFN Gate/Up | INT8 | **INT4** | **INT4** |
> > > | **Layer 63** | Attn / FFN Comm | **INT4** | INT4 / INT8 | **INT8** |
> > >
> > > **Key Takeaways from the allocation:** The uniform W8A8C4 naively compresses *all* communication to 4-bit, causing severe accuracy degradation. In contrast, CoCoQuant automatically learns to:
> > > 1. **Protect sensitive boundaries:** It allocates high precision (FP8) to highly sensitive modules (also shown in Fig.3) like `FFN Down`, and protects the critical communication at early (Layer 0) and late (Layer 63) stages with INT8.
> > > 2. **Aggressively compress robust layers:** To offset the latency cost of protecting those sensitive layers, it safely compresses the computation (e.g., `FFN Gate/Up`) and communication in middle layers (e.g., Layer 44) down to INT4.
> > >
> > > > ## 2. **Latency and Accuracy Comparison against vLLM SP W8A8C4:**
> > >
> > > To directly address your second question, we evaluated the uniform `vLLM SP W8A8C4` baseline. It achieves 7.54 PPL with 2704 ms TTFT. In contrast, our standard CoCoQuant configuration (from Table 2) prioritizes accuracy (6.92 PPL) with a TTFT of 2890 ms.
> > >
> > > Because CoCoQuant takes latency as a controllable budget constraint (Eq. 11), we can easily target a lower latency regime. By tightening the budget constraint, our solver yields a new configuration, `CoCoQuant*`:
> > >
> > > | Method | PPL | TTFT (ms) |
> > > |---|---:|---:|
> > > | vLLM + SP + W8A8C8 | 6.98 | 3768 |
> > > | vLLM + SP + W8A8C4 | 7.54 | 2704 |
> > > | **CoCoQuant** (Table 2) | **6.92** | **2890** |
> > > | **CoCoQuant$^*$** (new) | **7.34** | **2671** |
> > >
> > >
> > > As shown above, compared to the uniform `vLLM SP W8A8C4`:
> > > - `CoCoQuant` (from our paper) matches the uncompressed baseline's accuracy (6.92 PPL) while operating significantly faster than W8A8C8.
> > > - `CoCoQuant*` demonstrates a strictly superior Pareto frontier compared to the uniform W8A8C4: it achieves **both lower latency (2671 ms vs. 2704 ms) and better accuracy (7.34 vs. 7.54 PPL).**
> > >
> > > In practice, we can always trade off accuracy for latency by adjusting the latency budget. and CoCoQuant ensure the allocation results is always optimal under the given latency budget.
> > >
> > > ---
> > >
> > > We sincerely appreciate this follow-up. It prompted a much deeper analysis of our allocation strategy, and incorporating this case study into the revision will significantly help readers understand the inner workings of CoCoQuant!

---

### Official Review · Reviewer_CKyC · 2026-03-13

**Soundness:** 2
**Presentation:** 3
**Significance:** 3
**Originality:** 2
**Overall Recommendation:** 4
**Confidence:** 3

**Summary:**

This paper studies quantization for tensor-parallel LLM inference when communication becomes a major bottleneck. The main idea is that prior work often handles computation quantization and communication compression separately, which leads to repeated quantize/dequantize steps and inefficient precision flow. To address this, the paper proposes CoCoQuant, which combines three components: graph rewriting that moves row-wise operators such as RMSNorm and MoE gating before AllGather when possible, hardware-aware mixed-precision bitwidth allocation using ILP, and fused kernels for the rewritten local pipeline. Experiments on several dense and MoE models show improved latency-accuracy tradeoffs compared to full precision and simple uniform quantization baselines.

**Compliance With Llm Reviewing Policy:**

Affirmed.

**Final Justification:**

Novelty is limited, but the paper demonstrates strong engineering quality and solid execution

**Key Questions For Authors:**

1. Can the authors provide more direct evidence that the proposed mixed-precision allocation objective is effective in practice? While the paper shows that mixed allocation is better than uniform settings under matched latency budgets, I would still like to better understand why the proxy used in the allocation leads to good final policies.
2. Can the authors clarify how the reported speedups are measured? It would be helpful to better understand the runtime setup, including whether the fused kernels are integrated into the end-to-end system and how the batch/sequence configurations are chosen for the different experiments.

**Limitations:**

No. The paper should discuss the limitations of the proposed work.

**Strengths And Weaknesses:**

**Strengths**

The paper addresses an important practical problem in tensor-parallel LLM inference, where communication can become a major bottleneck. The overall motivation is reasonable, and the paper is generally easy to follow. In particular, the main intuition in Figure 2 is clear. I also think the hardware-aware mixed-precision allocation makes the work more substantial than a single graph optimization. The experiments include both dense and MoE models, and the results suggest that the proposed method can achieve a better latency-accuracy tradeoff than simple uniform low-bit settings in several cases.

**Weaknesses**

The graph rewrite is useful, but the core idea felt somewhat straightforward, and overall this part came across more as a careful systems optimization than a fundamentally new contribution. The mixed-precision allocation is one of the stronger parts of the paper, and the paper does show that mixed allocation performs better than uniform settings under matched latency budgets. Still, I would have liked a bit more analysis explaining why the proposed allocation objective is effective in practice.

---

> ### Author Rebuttal · Authors · 2026-03-30
>
> We greatly appreciate the reviewer's recognition of our motivation and of the mixed-precision component. We address each point below. Supporting tables/figures referenced below are collected in this anonymous PDF: https://1drv.ms/b/c/316d36085eac4395/IQDWtM23weHmT57t2AprhCsZAeBMxYC0yQN8AhHX1Hj-zhM?e=d4mkOb
>
> > ## 1. **W1: Graph rewrite novelty**
>
> We agree that graph rewriting, viewed purely as a topological transformation, is a known systems mechanism. Our intended claim is therefore narrower: the novelty is not operator movement itself, but that this rewrite becomes an indispensable precision-alignment mechanism in the entirely new context of quantized communication.
>
> Without this rewrite, placing high-precision row-wise operators (like RMSNorm or MoE routers) between a quantized AllGather and a quantized GEMM forces the system to insert explicit, redundant Dequantize/Quantize (Q/DQ) casting operations. This "precision fragmentation" completely wipes out the latency benefits of communication compression.
>
> Our rewrite physically aligns the precision flow to eliminate these high-precision bubbles. Furthermore, as empirically shown in Table 3, this quantization-aware alignment is not merely a system speedup trick; it is also critical for preventing catastrophic accuracy collapse. A concrete systems ablation is also provided in the **anonymous PDF (Section A)**, where `CoCoQuant w/o Fused` shows the large Q/DQ overhead without precision-aligned fusion.
>
> For a deeper discussion on this—including how our objective fundamentally diverges from standard Sequence Parallelism (e.g., handling MoE topk_idx metadata)—please refer to **our Response to Reviewer WYpS (Common Response 1, CR1)**.
>
> > ## 2. **W1,Q1: Effectiveness and Robustness of the Allocation Proxy (CR3 - Common Response 3)**
>
> We agree that the proxy's mechanism should be explained more clearly. Our calibration objective minimizing the relative Frobenius norm of intermediate outputs serves as a scale-invariant ranking signal for bit allocation, and is highly effective for three reasons:
>
> 1. **Alignment with Established PTQ Principles:** Our choice builds on a broad line of post-training quantization (PTQ) research demonstrating that local output perturbation is a practical and reliable surrogate for task loss. Foundational works such as Choukroun et al. (2019), AdaRound (Nagel et al., 2020), and BRECQ (Li et al., 2021) show that local fidelity objectives provide a practical compromise between global accuracy modeling and optimization tractability.
>
> 2. **Unified Metric for a Heterogeneous Space:** We fully acknowledge that other metrics: Hessian/Fisher-based sensitivitiy metrics are also principled. Our choice is not because second-order metrics are invalid. Rather, our setting requires jointly allocating precision over a mixed design space of both computation and communication collectives. A relative activation-perturbation metric provides a natural "common denominator" to score all candidate modules uniformly under one criterion.
>
> 3. **The Necessity of the Relative Formulation:** If we used absolute Frobenius error, modules with naturally large activation magnitudes would unjustly dominate the ILP budget. The normalized form ensures the solver compares the true proportional sensitivity across layers, preventing high-energy but robust layers from skewing the allocation.
>
> **Empirical Evidence:** The paper already provides three strong empirical proofs:
>
> - Figure 3 shows extreme heterogeneity across layers/modules, directly motivating mixed-precision allocation.
>
> - Table 4 demonstrates our ILP-selected policies substantially outperform uniform ones under the same latency budget (e.g., Llama3-8B-Instruct accuracy recovers from a collapsed 46.63% to 73.62%).
>
> - Table 2 proves these policies do not appear to overfit to a single model architecture.
>
> In revision, we will cite these foundational works to position our proxy more explicitly. We will also add a case study visualizing the learned bit allocations to concretely show how the proxy protects sensitive modules while safely compressing robust ones. Thanks for your insightful feedback!
>
> > ## 3. **Q2: How the reported speedups are measured**
>
> The fused kernels are fully integrated into our end-to-end testing system (VLLM TP offline inference engine). For the main results (Table 2), all experiments were conducted under a fixed setting of **Batch Size = 4** and **Sequence Length = 8K**. Furthermore, we comprehensively map out performance scaling across different **Batch Size × Sequence Length** configurations in Appendix Figure 4.
>
> - We rigorously discuss the baseline fairness in **CR2**. Due to space limitation, please refer to our **response to Reviewer dFpG (Common Response 2, CR2)**.
> - We also provide a comprehensive latency breakdown and comparison with more competitive baselines in **CR4**. Please refer to our **response to Reviewer ynLp (Common Response 4, CR4)** and the **anonymous PDF (Section A)**.

---

> > ### Author Rebuttal · Reviewer_CKyC · 2026-04-03
> >
> > Thank you for the response. My concerns have been adequately addressed. I will increase my score.

---

### Official Review · Reviewer_dFpG · 2026-03-13

**Soundness:** 2
**Presentation:** 2
**Significance:** 3
**Originality:** 3
**Overall Recommendation:** 4
**Confidence:** 2

**Summary:**

This paper argues that, in tensor-parallel (TP) inference, communication increasingly dominates end-to-end latency, so quantizing computation alone does not overcome the bandwidth wall. The authors propose CoCoQuant, which jointly optimizes communication and computation quantization to avoid a fragmented precision flow. In this setting, fragmentation leads to repeated quantization and dequantization, fewer fusion opportunities, and compounded quantization error. CoCoQuant combines precision-aligned graph rewriting that moves selected per-token or per-row operators across communication boundaries, a hardware-aware mixed-precision assignment that uses an error proxy from a small calibration set together with a latency model to solve an integer linear program (ILP) for per-module quantization under budget constraints, and fused kernels that realize these design choices as wall-clock speedups. Experiments report substantial gains in bandwidth-constrained settings with near-lossless quality, and the method appears more stable than uniform bit-width quantization or naïve composition baselines. The paper concludes that breaking the bandwidth wall requires treating communication and computation as a single co-design problem.

**Compliance With Llm Reviewing Policy:**

Affirmed.

**Final Justification:**

Thanks for the response from authors. I tend to keep my score.

**Key Questions For Authors:**

The graph-rewriting claim would benefit from a clearer correctness statement. You state that certain per-row or per-token operators can commute with AllGather, but the paper does not spell out the exact operator class covered by PGR or the conditions under which the rewrite is semantics-preserving.

The discussion of error attribution is also underspecified. The paper argues that a fragmented precision flow accumulates error and that fewer quantization and dequantization steps improve quality, yet CoCoQuant still introduces quantization error in both communication and computation, and it largely changes where quantization occurs through alignment and fusion. The paper should identify which mechanism explains the observed quality gains, whether it is the reduced number of quantize and dequantize steps, the placement of sensitive operators after higher-precision accumulation, or the bit-width choices produced by the ILP.

The speedups may similarly be driven by kernel engineering. It is plausible that more aggressive fused kernels and fewer memory round trips account for a large part of the wall-clock gains, independent of the proposed co-design. To support the main claim, the paper should separate the contributions of communication quantization, compute quantization, graph rewriting, and fusion. In particular, it would be useful to report how much of the speedup remains if the same fused kernels are applied to strong baselines without the ILP or rewriting.

The ablation study does not yet isolate the decisive components. Table3 supports PGR and Table4 supports mixed-precision optimization, but cross-ablation settings are missing, for example ILP without fusion, or precision alignment paired with a heuristic mixed-precision policy. These additional comparisons would make the causal explanation for the gains much clearer.

**Limitations:**

CoCoQuant is explicitly designed for Tensor Parallelism (TP) and relies heavily on the commutative property of row-wise operations with respect to AllGather. It does not extend to other parallelism paradigms commonly used in distributed inference, such as Pipeline Parallelism (PP) or Data Parallelism (DP) , where communication patterns (e.g., point-to-point sends/receives, all-reduce) lack the same algebraic properties. This restricts the framework's generalizability to diverse deployment scenarios.

The experiments focus almost exclusively on the prefill (TTFT) phase of inference. However, in many real-world LLM serving scenarios (e.g., chatbots, code completion), the decoding phase dominates latency due to memory-bound operations and repeated communication. CoCoQuant does not evaluate its effectiveness in the decoding phase, where communication patterns and quantization sensitivity may differ significantly. This leaves a critical gap in understanding its end-to-end practical impact.

**Strengths And Weaknesses:**

## Strengths
1. The problem formulation is well motivated. When communication dominates end-to-end latency, compute-only quantization naturally plateaus, and the paper attributes this behavior to a system-level structural issue.
2. For the framework, the paper frames communication compression and compute quantization as a single end-to-end design space and it focuses on precision-flow alignment instead of isolated per-operator choices. This system perspective is useful. The method also relies on the equivalence $f({AllGather}(x)) \equiv {AllGather}(f(x))$ to motivate graph rewrites that reduce redundant quantization and dequantization, and the rationale is straightforward.
3. On the experimental side, the evaluation spans multiple models, including MoE variants, and multiple tasks reported with zero-shot metrics and perplexity, and it includes ablations of key components such as precision-aligned graph rewriting (PGR), mixed-precision optimization, and fused kernels.
## Weaknesses
1. The latency cost model is simplified and may lead to suboptimal configurations in practice. Use an idealized upper-bound model to estimate latency, and then apply a few fixed scaling factors to approximate the real throughput that does not reflect several practical aspects of inference, such as compute-communication overlap, collective contention, and how performance changes with batch size and sequence length. When these effects cause noticeable prediction error, the ILP solution that is optimal for the proxy cost can deviate from the configuration that minimizes wall-clock latency.
2. The end-to-end gains may be dominated by fused-kernel engineering, and the paper does not yet disentangle this effect from the proposed co-design. The appendix micro-benchmarks report a wide gap between fused and unfused implementations, and the magnitude of this gap depends on whether the compiler manages to fuse the relevant operators. This sensitivity suggests that the reported speedups may depend strongly on the particular implementation stack, which weakens the argument that co-design is the primary source of improvement.
3. Although the paper cites FlashComm, Flux, and TileLink, it does not clearly position itself against end-to-end tensor-parallel inference runtimes and compilers that optimize communication-computation overlap and operator fusion, or against quantization-aware compilation and scheduling.  Without these comparisons, it is hard to judge whether the framework offers a distinct capability or largely overlaps with existing system optimizations.

---

> ### Author Rebuttal · Authors · 2026-03-30
>
> We thank the reviewer for highlighting the usefulness of our system perspective and our comprehensive evaluation on vary models. Supporting tables referenced below are collected in this anonymous link: https://1drv.ms/b/c/316d36085eac4395/IQDWtM23weHmT57t2AprhCsZAeBMxYC0yQN8AhHX1Hj-zhM?e=d4mkOb
> > ## 1. **W1: Latency Cost Model Idealization**
>
> We agree our cost model abstracts runtime complexities, but it is highly accurate for prefill. With large token counts, operators saturate compute, interconnect, and memory bandwidths, making FLOPs/memory traffic-based latency estimation reliable.
>
> Crucially, our model is input-shape-aware. While dynamic overlapping affects absolute wall-clock time, such system dynamics are orthogonal to the relative cost-benefit analysis needed for bit-width allocation. We validate estimated vs. actual latencies in **anonymous PDF, Sec B**, empirical validations show our model predicts actual latencies with exceptional precision for dense models (<2% error on Llama3.3-70B) and robust relative fidelity for complex MoEs (<19% error on Scout-109B), fully validating its reliability for guiding our ILP search.
> > ## 2. **W2,Q3: Baseline Fairness and Implementation Details (CR2 - Common Response 2)**
>
> We apologize for the confusion regarding our evaluation setup. Our evaluation is strictly fair and **built directly on top of vLLM (version 0.11.2.dev278, commit dc837bc23e)**. Since vLLM lacks native W4A4 support, we implemented CUTLASS per-token/per-channel GEMMs, integrating optimized ATOM (Zhao et al., MLSys 2024) for W4A4g128.
>
> Crucially, **we equipped all communication-quantized baselines (e.g., uniform W4A4C4 in Table 2) with the exact same custom fused kernels developed for CoCoQuant (e.g., `dequant+reduce+rmsnorm+residual+quant`)**. The speedups we report are **not** the result of comparing an optimized CoCoQuant against a naive, unfused implementation. The gains stem entirely from CoCoQuant's co-designed computation-communication co-quantization.
>
> - To explicitly demonstrate the speedup source, please refer to the **anonymous PDF (Section A)** and our **response to Reviewer ynLp (CR4)** for the finer-grained latency breakdown.
>
> > ## 3. **W3: Position of CoCoQuant Compared to FlashComm/Flux/TileLink etc.**
>
> - **FlashComm** provides efficient **quantized collective primitives**.
> - **Flux / TileLink** provide **runtime support for compute-communication overlap** without involving communication compression.
> - **CoCoQuant** belongs to the pareto optimal methods achieving communication-computation co-quantization with system design. At algorithm level, it dictates which computation and communication modules should use which precisions with strict precision alignment enforcement while at system level, it provides optimized kernel to achieve real-world latency reduction. Computation-communication overlap is not included in this work.
>
> This distinction matters because comm compression and overlap alone do not answer the main question: **once communication is quantized, how should the computation be jointly optimized so that whole inference pipeline achieves pareto optimal latency and accuracy?** In this sense, Flux/TileLink can be viewed as orthogonal building blocks that can be adopted by CoCoQuant to achieve better performance (e.g. fuse GEMM with A2A, which is decomposed from AllReduce). While FlashComm is a good choice for communication-only compression, CoCoQuant provides a better practical, holistic solution for communication-computation co-quantization.
> > ## 4. **Q1: Graph Rewriting Correctness**
>
> The rewrite $f(\text{AllGather}(x)) = \text{AllGather}(f(x))$ is strictly semantics-preserving iff **$f(\cdot)$ operates independently on the token dimension**. We will explicitly define this in Section 3.1.
>
> In mainstream LLMs, the operations between TP reduction and the next GEMM strictly belong to this **token-independent** class: row-wise norm, element-wise activations, and token-wise routing. Applying them to local sequence shards before `AllGather` is mathematically identical to applying them globally after.
>
> **Counter-example:** It is invalid for operations introducing inter-token dependencies, like RWKV's Token Shift (which blends adjacent tokens before linear projection). Applying this shift on sharded sequences before AG breaks semantic equivalence.
> > ## 5. **Q2,Q4: Accuracy Performance Gain Source, Error Attribution and Cross-Ablation**
>
> To isolate accuracy gains, we add a cross-ablation study on Scout-109B **(see anonymous PDF, Sec D)**. The breakdown proves CoCoQuant's accuracy does not come from a single "magic" trick: Moving the MoE router before AG preserves unquantized routing semantics. Fusing operators eliminates intermediate ping-pong Q/DQ casts, removing massive truncation noise. Finally, our ILP solver perfectly balances remaining error sensitivity.
> > ## 6. **Decoding Limitation**
> please refer to **our response to Reviewer WYpS (CR5)** for the detailed discussion.

---

> > ### Author Rebuttal · Reviewer_dFpG · 2026-03-31
> >
> > Thanks for the response from authors. I tend to keep my score.

---

### Official Review · Reviewer_WYpS · 2026-03-15

**Soundness:** 3
**Presentation:** 3
**Significance:** 3
**Originality:** 3
**Overall Recommendation:** 4
**Confidence:** 5

**Summary:**

CoCoQuant combines precision-aligned graph rewriting to reduce redundant Q/DQ overhead, fused communication/computation kernels to improve efficiency across collective boundaries, and hardware-aware mixed-precision bit allocation via ILP to select high-performance quantization strategies while preserving accuracy.

**Compliance With Llm Reviewing Policy:**

Affirmed.

**Final Justification:**

The paper makes a meaningful contribution on hardware-aware ILP for per-module comm+compute precision allocation, plus precision-aligned fused execution for prefill-heavy TP serving. The rebuttal clarified the scope and strengthened the empirical case.

The rebuttal resolved a substantial part of my fairness concern. The authors clarified that the communication-quantized baselines use the same fused boundary kernels as CoCoQuant, and added a direct vLLM+SP quantized baseline showing that the benefit comes from the allocation/co-design rather than an unfair kernel comparison. The authors also clearly acknowledged that the method depends on the A2A + local reduce + AG decomposition and is mainly useful for prefill / TTFT, not decode-heavy serving.

**Key Questions For Authors:**

- How does the method behave in decode-heavy serving? The current motivation and evaluation appear largely prefill-oriented; could the authors estimate or evaluate whether the proposed communication/computation co-optimization remains beneficial when per-step communication becomes more latency-bound?
- How robust is the calibration proxy across tasks? Since the bitwidth allocation is guided by calibration error, can the authors clarify how well this proxy predicts end-task quality for workloads with different sensitivity to numerical perturbations?
- Is the method fundamentally tied to the A2A, local reduce, AG decomposition of quantized AllReduce, or can similar benefits also be realized in native RS/AR-based TP implementations? In particular, how does CoCoQuant compare against strong vLLM or SGLang baselines that already fuse reduce-scatter/all-reduce with post-reduction norm and quantization?

**Limitations:**

Yes

**Strengths And Weaknesses:**

**Strengths**
- The paper targets a practically important TTFT bottleneck: communication, particularly repeated Q/DQ around collective operations, is identified as a concrete source of inefficiency.
- Rather than treating communication quantization as an isolated add-on, the paper formulates communication, computation, and operator placement as a unified optimization space, which is a sensible and practically relevant perspective.
- The evaluation demonstrates significant end-to-end speedups with minimal perplexity degradation across diverse modern models, including both dense and large-scale MoE architectures.

**Weaknesses**
- The proposed graph rewriting closely resembles existing sequence-parallel optimizations in frameworks such as vLLM and SGLang. Pushing row-wise operations toward the sharded phase is already an established systems practice, which limits the conceptual novelty of this component;
- The TP baseline may be too conservative. It is unclear whether modern inference frameworks genuinely realize this naive five-step “ping-pong” Q/DQ flow in practice;
- The framework appears tightly coupled to a specific decomposition of AllReduce into A2A, local reduce, and AG, with the fused kernel built around DQ, Reduce, RMSNorm/Add/Route, and Q. It remains unclear how broadly the gains transfer to more standard RS/AR-centric implementations;
- The EP scope is less clearly established. Some EP implementations already support native FP8/FP4 communication with scale metadata, so the need for explicit Q/DQ boundaries is system- and kernel-dependent.

---

> ### Author Rebuttal · Authors · 2026-03-30
>
> We thank the reviewer for recognizing the practical importance of addressing the bandwidth wall. Supporting tables/figures referenced below are in this anonymous link: https://1drv.ms/b/c/316d36085eac4395/IQDWtM23weHmT57t2AprhCsZAeBMxYC0yQN8AhHX1Hj-zhM?e=d4mkOb
> > ## 1. **W1: Novelty of PGR vs. SP in Megatron/vLLM/SGLang (CR1 - Common Response 1)**
>
> We agree that PGR formally relates to Sequence Parallelism in Megatron: both move certain operators to the sharded phase. However, **PGR is not a reinvention of SP; it solves a fundamentally different problem unique to comm-comp co-quantization**. Standard SP moves operations primarily to reduce computation or memory footprint whereas PGR acts as a precision-alignment mechanism.
>
> - **PGR is Mandatory:** Consider a quantized `AllGather->GEMM`. **Even if both are assigned the same bit-width, the precision flow remains fragmented**: Row-wise operations (like Norm or MoE Gate) between them require FP32/BF16 computation, forcing explicit dequantize-requantize (DQ/Q) casts. PGR **mandates** moving these row-wise operations to sharded phase so that the output of AG can be directly consumed by GEMM, no more extra DQ/Q casts.
> - **The MoE Counter-Example:** The difference in objective leads to architectural changes. For instance, standard SP never moves MoE routing before AG as doing so alters the semantic content of the AllGather payload with negligible performance gain. For CoCoQuant, this is mandatory. To achieve this, we not only developed a fused `DQ->Reduce->Gate->Q` kernel, but we also developed a specialized AG kernel that piggybacks the `topk_idx` and scale factors alongside token activations to eliminate extra memory copies. This precision-driven architectural change fundamentally separates PGR from traditional SP.
> - **Empirical Impact:** Table 3 proves this quantization-aware rewrite prevents catastrophic accuracy collapse. We will clarify that PGR's novelty lies not in operator movement, but in being the indispensable mechanism making co-quantization executable.
>
> > ## 2. **W2: Baseline Fairness**
>
> Standard vLLM/SGLang lack TP quantization. Without PGR and custom kernels, precision-conversion bubbles (5 Q/DQs in Fig. 2) are unavoidable.
>
> Crucially, our baseline is not a naive unfused implementation; it uses the exact same custom fused kernels as CoCoQuant. Due to space limits, please refer to **CR2 (Response to Reviewer dFpG)** regarding baseline fairness, and the **anonymous PDF (Section A; see also CR4) for latency breakdowns against more vLLM baselines**.
> > ## 3. **W3,Q3: Native RS/AR Applicability**
>
> Our design relies on the `A2A+local_reduce+AG` decomposition because it explicitly exposes a local reduction phase for fusing post-reduction row-wise operators before the final gather. In native RS/AR (e.g., Ring AllReduce), reduction and transfer are interleaved; prior works such as Zero++ (Wang et al., 2023) show that directly quantizing these interleaved steps can introduce substantial Q/DQ overheads. We will explicitly acknowledge this implementation dependency in the revision.
> > ## 4. **W4: EP Scope and Native FP8/FP4**
>
> As stated in Sec 2.2 (line 133), we prioritize TP for now. EP compression is out of scope. However, we'd like to discuss that CoCoQuant's principles remain conceptually necessary for EP. While DeepEP support native FP8/FP4 transfers, this only solves the network step. If surrounding expert use different precisions or quantization granularities, explicit Q/DQ cast problem inevitably reappear at boundaries. Our methodology: strictly aligned precision flow will be essential to eliminate this problem.
> > ## 5. **Q1: Decode-Heavy Serving Performance (CR5 - Common Response 5)**
>
> We agree; communication in decode-heavy scenarios are typically latency-bound, not bandwidth-bound. Preliminary evaluating Time-Per-Output-Token (TPOT) on Qwen-32B (BS=8, 8K KV length): **vLLM BF16 TP achieves 19.28 ms, while CoCoQuant$^1$ takes 26.74 ms**. This is expected: decode steps have tiny communication payloads, and vLLM use one-shot custom low-latency AllReduce kernel to optimize for this scenario. In contrast, CoCoQuant strictly relies on the `A2A+Reduce+AG` decomposition. The multi-step kernel launch and synchronization overheads of this decomposition simply outweigh the bandwidth savings.
>
> However, we want to clarify that Prefill-Decode Disaggregation has become an established paradigm in serving, allowing prefill and decode phases to utilize different deployment strategies. It's practical to deploy CoCoQuant exclusively on the prefill instance to reduce TTFT. Meanwhile, the decode cluster can retain existing low-latency optimizations. We will explicitly clarify this deployment scope.
> > ## 6. **Q2: Calibration Proxy Robustness**
>
> Due to space limitation, please see our response to **Reviewer CKYC (Point 1)** regarding the theoretical and empirical robustness of our proxy.

---

> > ### Author Rebuttal · Reviewer_WYpS · 2026-04-05
> >
> > My concerns have been adequately addressed. I'll increase score. Thanks.

---

### Decision · Program_Chairs · 2026-04-30

**Decision:**

Accept (regular)

**Comment:**

The authors have studied tensor-parallel inference and proposed CoCoQuant to jointly optimize communication and computation quantization. The key contributions are: precision-aligned graph rewriting that moves selected per-token or per-row operators across communication boundaries, hardware-aware mixed-precision bitwidth allocation using integer linear program, and fused kernels to realize wall-clock speedups. The authors have provided experiments for several dense and MoE models and show improved latency-accuracy tradeoffs compared to full precision and simple uniform quantization baselines.

The reviewers had concerns regarding the proper kernel comparison, lower latency in GEMM and communication compared to the vLLM SP W8A8C8 baseline, and novelty. After rebuttal and discussion, all reviewers are happy with the paper and noted the rebuttal addressed their major concerns. Considering the rebuttal and reviewers’ final justification, I would like to recommend acceptance and suggest that authors address all revisions that are promised within the camera-ready version.